# CogMath: Assessing LLMs' Authentic Mathematical Ability from a Human Cognitive Perspective

**Jiayu Liu** [1 2]   **Zhenya Huang** [2 3]   **Wei Dai** [1]   **Cheng Cheng** [2 3]   **Jinze Wu** [2 4]   **Jing Sha** [2 4]   **Song Li** [2 4]   **Qi Liu** [1 2]
**Shijin Wang** [2 4]   **Enhong Chen** [2 3]

## Abstract

Although large language models (LLMs) show promise in solving complex mathematical tasks, existing evaluation paradigms rely solely on a coarse measure of overall answer accuracy, which are insufficient for assessing their authentic capabilities. In this paper, we propose **CogMath**, which comprehensively assesses LLMs' mathematical abilities through the lens of human cognition. Specifically, inspired by psychological theories, CogMath formalizes human reasoning process into 3 stages: *problem comprehension*, *problem solving*, and *solution summarization*. Within these stages, we investigate perspectives such as numerical calculation, knowledge, and counterfactuals, and design a total of 9 fine-grained evaluation dimensions. In each dimension, we develop an "*Inquiry-Judge-Reference*" multi-agent system to generate inquiries that assess LLMs' mastery from this dimension. An LLM is considered to truly master a problem only when excelling in all inquiries from the 9 dimensions. By applying CogMath on three benchmarks, we reveal that the mathematical capabilities of 7 mainstream LLMs are overestimated by 30%-40%. Moreover, we locate their strengths and weaknesses across specific stages/dimensions, offering in-depth insights to further enhance their reasoning abilities.

## 1. Introduction

The rise of large language models (LLMs) has marked a pivotal moment in artificial intelligence. Particularly within the realm of mathematical reasoning, these models have made breakthroughs in solving complex mathematical problems (Wei et al., 2022; Xue et al., 2024). For example, GPT-4 has achieved over 75% accuracy on the high school competition-level MATH dataset (Hendrycks et al., 2021). More recently, the OpenAI-o1 model has surpassed 70% accuracy on the AIME math competition, placing it at a level comparable to the top 500 US high school students [1]. This remarkable progress has not only redefined the potential of AI in mathematics but also spurred a growing body of research dedicated to evaluating and understanding the mathematical proficiency of these models.

To assess the mathematical ability of LLMs, numerous benchmarks have been proposed (Li et al., 2024a). For instance, E-GSM (Xu et al., 2025) introduces problems across four length ranges to assess LLMs' generalization with respect to input length. GSM-Plus (Li et al., 2024b) introduces eight variants of GSM8K dataset (Cobbe et al., 2021) to investigate the robustness of LLMs. MPA (Zhu et al., 2024) rewrites existing datasets based on five principles to address the effects of data contamination. However, on one hand, some benchmarks are overly task-specific, focusing on narrow aspects of reasoning (e.g., long-text understanding) by designing specialized problem types (e.g., long-text questions). On the other hand, they rely on a coarse accuracy metric that overlooks the detailed reasoning processes. As a result, they are unable to fully grasp the entire spectrum of mathematical capabilities that LLMs possess.

In this paper, we propose **CogMath**, which offers a scientific and comprehensive evaluation of LLMs' mathematical abilities by delving into the cognitive stages of human reasoning processes. Specifically, psychological research points out that human mathematical reasoning typically involves three stages (Schoenfeld, 2014; Lesh & Doerr, 2003; Dehaene et al., 1999): *problem comprehension*, *problem solving*, and *solution summarization*. Aligned with these stages, we design nine evaluation dimensions covering key aspects such as computation, knowledge, and counterfactual reasoning. For example, in *problem comprehension* stage, we assess

[1]School of Artificial Intelligence and Data Science, University of Science and Technology of China [2]State Key Laboratory of Cognitive Intelligence [3]School of Computer Science and Technology, University of Science and Technology of China [4]iFLYTEK AI Research. Correspondence to: Enhong Chen <cheneh@ustc.edu.cn>.

*Proceedings of the 42nd International Conference on Machine Learning*, Vancouver, Canada. PMLR 267, 2025. Copyright 2025 by the author(s).

---

[1]https://openai.com/index/openai-o1-mini-advancing-cost-efficient-reasoning/

the model's ability to handle different formulations of the same problem. In *problem solving* stage, we break down the solution into three orthogonal components: problem-solving strategy, numerical computation, and knowledge application, and evaluate LLMs in each aspect independently. In *solution summarization* stage, we go beyond traditional forward evaluation by introducing intermediate-step questions and backward reasoning tasks, testing whether the model can trace back through its reasoning pathway.

To carry out the evaluation in each dimension, we also design an "*Inquiry-Judge-Reference*" multi-agent system: the *Inquiry* agent poses a dimension-specific inquiry about the problem, the *Judge* agent refines the inquiry to ensure its quality, and the *Reference* agent provides a correct answer as a standard to assess the LLM's performance. Unlike traditional evaluation paradigms that rely solely on an answer accuracy, CogMath considers an LLM to truly master a problem only after excelling in all inquiries in 9 dimensions.

We apply CogMath on the most representative mathematical benchmarks GSM8K and MATH, along with an additional dataset we collected, MExam, which is composed of real exam tests that cover the full K-12 curriculum. Then, we evaluate 7 mainstream LLMs including GPT-4 (Achiam et al., 2023), GPT-3.5-Turbo (OpenAI, 2023), Gemini-1.5-Flash (Team et al., 2023), DeepSeek-V2.5 (Liu et al., 2024a), Llama3-8B (Meta, 2024), Llama2-13B (Touvron et al., 2023), and Mixtral-8x7B-Instruct (MistralAITeam, 2023). Our key experimental findings are as follows [2]:

- The authentic mathematical capabilities of current LLMs are overestimated by 30%-40%. For instance, GPT-4 has truly mastered only 39.7% and 67.1% of the problems in MATH and GSM8K datasets, respectively. Moreover, this overestimation is not solely attributable to data contamination, but rather to an excessive imitation of superficial patterns of reasoning.

- We locate the deficiency stage of LLMs. Weaker models (e.g., Llama2-13B) still struggle in *problem comprehension* stage, while stronger models (e.g., DeepSeek-V2.5) face challenges primarily in *problem solving* stage, particularly in their mastery of knowledge.

- Confronted with a counterfactual setting, LLMs may exhibit an inherent "over-correction" behavior, automatically aligning with patterns from the training data.

- Existing prompting techniques, such as CoT and ICL, fail to truly improve the mathematical reasoning capabilities of LLMs.

---

[2]Our code and data are available at https://github.com/Ljyustc/CogMath.

## 2. Related Work

### 2.1. Large Language Models

Large language models (LLMs) have significantly advanced the field of natural language processing (NLP) (Min et al., 2023; Zhao et al., 2024a; Liu et al., 2025; Zhao et al., 2024b). Models like OpenAI-o1, GPT-4 (Achiam et al., 2023), and GPT-3.5-Turbo (OpenAI, 2023) have set new performance milestones across numerous NLP tasks, such as sentiment classification (Zhang et al., 2024c), question answering (Hendrycks et al., 2021), and translation (Wang et al., 2023a). To further enhance their reasoning and problem-solving abilities, several advanced techniques have been introduced. Among them, Chain-of-Thought (CoT) (Wei et al., 2022), Tree-of-Thought (ToT) (Yao et al., 2024), and Graph-of-Thought (GoT) (Besta et al., 2024) simulate structured and logical reasoning paths using chains, trees, and graphs, respectively, allowing models to handle multi-step problems more effectively. In-Context Learning (ICL) (Dong et al., 2022) enables a model to learn from a few examples to generalize and solve unseen problems. In addition, there are many other key techniques, such as self-consistency (Wang et al., 2023b), Retrieval-Augmented Generation (RAG) (Chen et al., 2024), and tool learning (Ma et al., 2025). We refer the readers to a more detailed survey conducted by (Zhao et al., 2023).

### 2.2. Evaluation on LLM Mathematical Ability

We categorize existing mathematical benchmarks from two perspectives: problem difficulty and problem types. In terms of difficulty, MATH (Hendrycks et al., 2021) and CHAMP (Mao et al.) are representative high school competition-level datasets, while GSM8K (Cobbe et al., 2021) and MAWPS (Koncel-Kedziorski et al., 2016) are elementary-level math word problems. From the perspective of problem types, E-GSM (Xu et al., 2025) includes four categories of math problems of varying lengths to evaluate LLMs' generalization on longer contexts, TheoremQA (Chen et al., 2023) and MathBench (Liu et al., 2024b) test LLMs' ability to prove and apply theorems, while MathVista (Lu et al., 2024) and GeoEval (Zhang et al., 2024b) focus on visual reasoning and geometric reasoning. To mitigate the impact of data contamination, some studies introduce perturbations into existing benchmarks, such as GSM-1k (Zhang et al., 2024a), GSM-Plus (Li et al., 2024b), and MPA (Zhu et al., 2024), which consist of manually-annotated/eight/five variations of GSM8K, respectively.

However, these works lack in-depth exploration of models' reasoning processes, instead relying on a coarse overall accuracy metric. This makes it difficult to precisely identify at which cognitive stage the LLM encounters issues and to provide further guidance for improving LLMs.

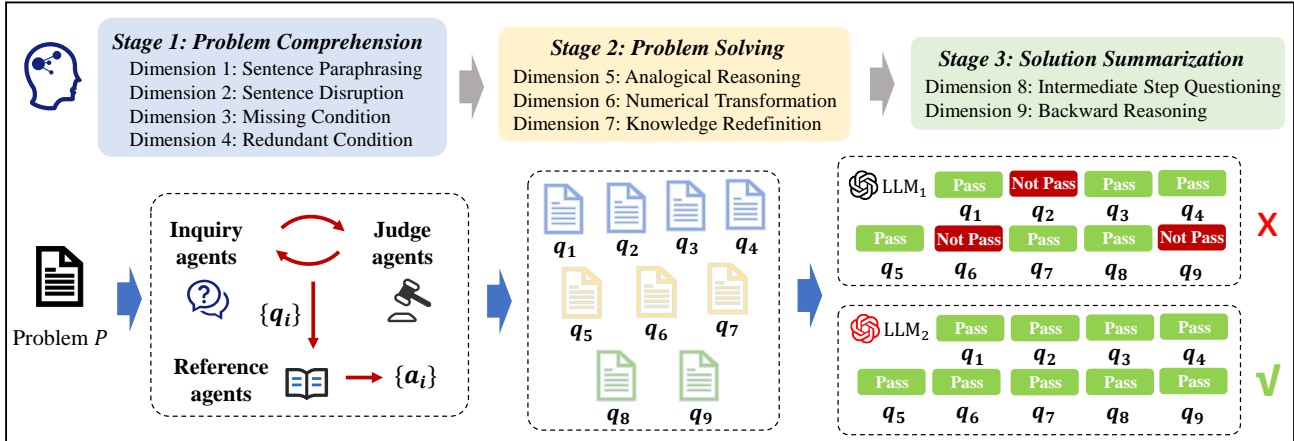

Figure 1: Illustration of our CogMath framework.

# 3. CogMath

To achieve a comprehensive evaluation, we draw inspiration from how humans solve mathematical problems. Specifically, psychological theories indicate that human reasoning process consists of three stages: *problem comprehension*, *problem solving*, and *solution summarization* (Schoenfeld, 2014; Lesh & Doerr, 2003; Dehaene et al., 1999). They build upon each other, with each stage taking the output of the previous one as input. *Problem comprehension* involves analyzing problem $P$'s information, such as word semantics, text structure, and given conditions. *Problem solving* stage combines the problem information with relevant knowledge to infer a solution. Finally, in *solution summarization* stage, humans engage in self-summarization, reviewing their thought processes, organizing logical steps, and forming a structured methodology.

Therefore, as illustrated in Figure 1, in our **CogMath** framework, we evaluate LLMs' mathematical abilities from the above three stages. For each stage, we design multiple dimensions to assess LLMs from various perspectives. For instance, to assess a model's *problem comprehension* stage, beyond investigating its performance after rephrasing the original problem, we can explore its sensitivity to changes in problem conditions, such as adding irrelevant information or disrupting problem sentences. Overall, for these three stages, we develop a total of nine dimensions that form a cohesive and comprehensive evaluation, with an example presented in Appendix A.

To quantify an LLM's performance in each dimension $i$, we design an "*Inquiry-Judge-Reference*" multi-agent system. As shown in Figure 1, the *Inquiry* agent poses an inquiry $q_i$ based on $P$ that aligns with dimension $i$. The *Judge* agent evaluates the quality of $q_i$ and repeatedly invokes the *Inquiry* agent until a reasonable inquiry is obtained or the maximum number of iterations $\delta$ is reached. The *Reference*

agent generates an answer $a_i$ to $q_i$, which is used to assess whether a real LLM's response to $q_i$ is correct. Prompts for all agents are presented in Appendix B. For humans, truly mastering a mathematical problem requires a solid performance at each dimension. Hence, in CogMath, only when an LLM passes all dimensions can we conclude that it has genuinely mastered the problem $P$. Notably, these evaluation results also serve as a multifaceted analysis of the model, revealing gaps between its performance in each dimension and human cognition.

## 3.1. Stage 1: *Problem Comprehension*

The *problem comprehension* stage involves capturing the details of words, phrases, and sentences in the problem, as well as translating the mathematical conditions on a broader scale. To assess an LLM in this stage, we design four dimensions from fine-grained to coarse-grained granularities:

• **Dimension 1: Sentence Paraphrasing.** If a human truly understands a mathematical problem, she will demonstrate a robust understanding of the problem's meaning despite changes in wording or sentence structure. Inspired by this, this dimension evaluates the LLM's ability to understand a problem that has been rephrased using synonymous expressions. To achieve this, we ask the *Inquiry* agent to pose a paraphrased version of the original problem $P$ as $q_1$, while preserving the mathematical essence (e.g., "Jacob had \$21 ..." in Table 6). Since the answer to the rephrased problem remains the same as the original, the *Reference* agent can directly use the original answer as the reference $a_1$.

• **Dimension 2: Sentence Disruption.** To prevent an LLM from simply memorizing a solution, we propose this dimension from a counterfactual perspective: the *Inquiry* agent randomly disrupts the word order within each clause of the original problem, creating a "pseudo problem" $q_2$, where the words remain the same as in $P$, but from a human per-

spective, $q_2$ is unreadable and unsolvable. In this case, the *Reference* agent does not need to generate an answer, as the expected response is simply "unsolvable", and the *Judge* agent is also no longer required to make any judgments. If an LLM's response to $q_2$ is the same with the original answer, it indicates that this model is likely recalling an answer based on certain keywords or patterns rather than truly understanding the problem. Therefore, this dimension helps us assess whether the LLM is genuinely solving the problem or relying on superficial clues (Sun et al., 2023).

● **Dimension 3: Missing Condition.** For humans, understanding what the given conditions are in a math problem is a critical step in the comprehension process. If essential conditions are missing, we can recognize that the problem becomes unsolvable. Therefore, in this dimension, we still adopt a counterfactual approach: if, after the removal of a necessary condition, the LLM is still able to produce the original answer, it suggests that the model is relying on the semantic similarity to the memorized problem to map out the solution, rather than genuinely solving it. As illustrated in Appendix B.2.1, we ask the *Inquiry* agent to omit one key condition from the original problem, presenting an underspecified version of $P$ as inquiry $q_3$. The *Judge* agent needs to carefully assess whether only one condition has been removed and whether the inquiry $q_3$ does not alter any other parts of the original problem's formulation.

● **Dimension 4: Redundant Condition.** In contrast to Missing Condition, this dimension introduces irrelevant conditions into the problem. For example, an extra condition such as "Before meeting ... make sure it was correct" might be added to the problem shown in Table 6. An LLM that truly masters problem $P$ should distinguish between essential and non-essential information, ensuring that unnecessary data does not interfere with the reasoning process. Therefore, the *Inquiry* agent presents a problem $q_4$ with one redundant condition. The *Judge* agent evaluates whether the extraneous detail does not affect the solution to the original problem, and the *Reference* agent provides the original answer, as the added information should not affect the solution.

### 3.2. Stage 2: *Problem Solving*

Problem solving requires three orthogonal components: solving strategy, numerical calculation, and mathematical knowledge (Sweller, 1988; Jonassen, 2000). The solving strategy is a logical organization specific to the problem, numerical calculation refers to arithmetic operations, and mathematical knowledge reflects common principles that apply across problems. To evaluate whether an LLM genuinely grasps them, we design the following three dimensions:

● **Dimension 5: Analogical Reasoning.** The solving strategy serves as a commonality across different problems, allowing a human to solve multiple similar problems using

the same underlying logic. In this dimension, the *Inquiry* agent presents a new problem that is conceptually consistent to, but not identical to, the original problem $P$, as $q_5$ (e.g., "Tom had 21 comic ..." in Table 6). This tests the LLM's ability to generalize the solving strategy, reflecting its grasp of the underlying reasoning thought. To be notice, $q_5$ preserves the original problem-solving process, maintaining the same approach, difficulty, and required knowledge, with no change to the core principles or complexity.

● **Dimension 6: Numerical Transformation.** Generally, the solving strategy represents the essential structure of solution, and the final solving step can be seen as plugging the numerical values from the original problem into the strategy. Therefore, if a human has mastered the problem, changing the numerical values will not affect the ability to solve it (Polya, 2014; Leighton & Gierl, 2007). Based on this idea, in this dimension, the inquiry $q_6$ is a variant of the original problem $P$ that modifies its numerical values (e.g., replace the numbers "21" and "100" with "30" and "120" in Table 6). Since $q_6$ is a new problem, we instruct the *Reference* agent to refer to the original answer and provide a corresponding new answer.

● **Dimension 7: Knowledge Redefinition.** Knowledge forms the foundation of human cognition, guiding how abstract principles are applied during the solution process (Goldman, 1986; Habermas, 2015; Liu et al., 2023; 2025). For example, solving a geometric problem may require the formula knowledge for triangle area. This understanding is flexible—if the problem redefines the formula of "triangle area", a human who truly grasps the concept will adapt her reasoning to fit the new definition.

To assess if an LLM can do this, the *Inquiry* agent modifies a key mathematical definition within problem $P$ by introducing a statement like "Assume the area formula of a triangle is defined as four times the sum of the lengths of its sides" in inquiry $q_7$. This redefinition forces the LLM to adapt its solution based on the modified concept. The *Reference* agent then generates a new solution based on the redefined knowledge, and the *Judge* agent assesses whether $q_7$ with the new definition is solvable.

### 3.3. Stage 3: *Solution Summarization*

After completing a problem-solving stage, humans often reflect on their reasoning process, summarizing the steps and the methodology behind their approach (Cottrell, 2023; Dewey, 2022). This summarization helps consolidate the understanding of not just the solution, but also the overall thought process, which can then be applied to similar problems in the future. In this stage, a human that truly masters the problem can accurately recall intermediate reasoning steps and verify the solution by working backward. To mimic these processes, we examine two critical dimensions:

| | | MATH | | | | | | | | GSM8K | MExam |
|---|---|---|---|---|---|---|---|---|---|---|---|
| | | Avg | Alg | Count | Geo | Itmd | Num | Pre-Alg | Pre-Cal | | |
| GPT-4 | Vanilla | 0.758 | 0.908 | 0.783 | 0.660 | 0.580 | 0.792 | 0.879 | 0.574 | 0.954 | 0.807 |
| | CogMath | 0.393 | 0.532 | 0.395 | 0.276 | 0.197 | 0.337 | 0.587 | 0.266 | 0.671 | 0.364 |
| | Δ | -0.365 | -0.376 | -0.388 | -0.384 | -0.383 | -0.455 | -0.292 | -0.308 | -0.283 | -0.440 |
| GPT-3.5 | Vanilla | 0.482 | 0.672 | 0.426 | 0.390 | 0.276 | 0.415 | 0.693 | 0.273 | 0.838 | 0.531 |
| | CogMath | 0.176 | 0.280 | 0.108 | 0.121 | 0.062 | 0.109 | 0.315 | 0.088 | 0.424 | 0.192 |
| | Δ | -0.306 | -0.392 | -0.318 | -0.269 | -0.214 | -0.306 | -0.378 | -0.185 | -0.414 | -0.339 |
| Gemini-1.5 | Vanilla | 0.615 | 0.812 | 0.535 | 0.489 | 0.423 | 0.555 | 0.781 | 0.479 | 0.922 | 0.739 |
| | CogMath | 0.291 | 0.428 | 0.247 | 0.173 | 0.142 | 0.206 | 0.455 | 0.205 | 0.500 | 0.338 |
| | Δ | -0.325 | -0.385 | -0.288 | -0.316 | -0.281 | -0.349 | -0.326 | -0.274 | -0.422 | -0.401 |
| Llama3-8B | Vanilla | 0.336 | 0.458 | 0.258 | 0.217 | 0.194 | 0.267 | 0.540 | 0.222 | 0.826 | 0.455 |
| | CogMath | 0.056 | 0.081 | 0.044 | 0.025 | 0.016 | 0.024 | 0.123 | 0.020 | 0.342 | 0.096 |
| | Δ | -0.280 | -0.377 | -0.214 | -0.192 | -0.178 | -0.243 | -0.417 | -0.202 | -0.484 | -0.359 |
| Llama2-13B | Vanilla | 0.106 | 0.142 | 0.080 | 0.073 | 0.051 | 0.074 | 0.196 | 0.059 | 0.446 | 0.267 |
| | CogMath | 0.008 | 0.013 | 0.004 | 0.004 | 0.003 | 0.001 | 0.016 | 0.004 | 0.064 | 0.024 |
| | Δ | -0.098 | -0.129 | -0.076 | -0.069 | -0.048 | -0.073 | -0.180 | -0.055 | -0.382 | -0.243 |
| Mixtral-8x7B | Vanilla | 0.374 | 0.495 | 0.306 | 0.278 | 0.238 | 0.265 | 0.529 | 0.339 | 0.575 | 0.506 |
| | CogMath | 0.092 | 0.147 | 0.053 | 0.058 | 0.037 | 0.028 | 0.165 | 0.079 | 0.212 | 0.133 |
| | Δ | -0.282 | -0.348 | -0.253 | -0.220 | -0.201 | -0.237 | -0.364 | -0.260 | -0.363 | -0.373 |
| DeepSeek-V2.5 | Vanilla | 0.747 | 0.915 | 0.730 | 0.597 | 0.548 | 0.780 | 0.870 | 0.625 | 0.951 | 0.855 |
| | CogMath | 0.368 | 0.519 | 0.346 | 0.284 | 0.207 | 0.285 | 0.526 | 0.233 | 0.646 | 0.342 |
| | Δ | -0.379 | -0.396 | -0.384 | -0.313 | -0.341 | -0.495 | -0.344 | -0.392 | -0.305 | -0.513 |

Table 1: Performance of different LLMs on vanilla datasets and our CogMath framework.

• **Dimension 8: Intermediate Step Questioning.** In human reasoning, breaking down the problem-solving process into smaller, manageable steps is essential for clarity and learning. Beyond evaluating the final answer, assessing whether an LLM has precisely understood the intermediate steps is an indispensable part of determining if it truly grasps the solution. Therefore, in this dimension, the *Inquiry* agent presents an inquiry $q_8$ that asks an LLM to explain one of the key intermediate steps during the problem-solving process (e.g., step 2 in Appendix B.7.1). This ensures that the model is not just arriving at a correct final answer by coincidence or pattern recognition, but is following a clear, logical sequence throughout the entire solution. Then, the *Judge* agent checks whether $q_8$ corresponds to a specific step, and the *Reference* agent generates an explanation for this step based on the original solution.

• **Dimension 9: Backward Reasoning.** Inspired by (Yu et al., 2024; Weng et al., 2023), backward reasoning is a crucial and challenging task. It refers to inferring missing information from the solution, mirroring how humans check their thought by retracing their reasoning to ensure there are no mistakes (Rips, 1994). Therefore, it can be used to evaluate whether LLMs maintain consistency and logical coherence from both directions—forward and backward. If a model truly understands the problem-solving process, it should be able to perform this reverse reasoning without contradictions.

For this purpose, our *Inquiry* agent formulates inquiry $q_9$ by masking a key numerical value from the original problem $P$ and requiring the model to infer the missing value based

on the original solution. The *Reference* agent directly takes the masked value as the answer $a_9$, and the *Judge* agent evaluates whether the masked problem, when combined with the original answer, remains solvable.

## 4. Evaluation

### 4.1. Experimental Setup

We evaluated seven mainstream LLMs, including four closed-source models: GPT-4 (Achiam et al., 2023), GPT-3.5-Turbo (OpenAI, 2023), Gemini-1.5-Flash (Team et al., 2023), and DeepSeek-V2.5 (Liu et al., 2024a), as well as three open-source models: Llama3-8B (Meta, 2024), Llama2-13B (Touvron et al., 2023), and Mixtral-8x7B-Instruct (MistralAITeam, 2023). The implementation details and problem sets of CogMath are described in Appendix C. We adopt *Pass Rate* (PR) as our metric. This is because, in CogMath, dimensions 2 and 3 are based on counterfactual settings. Therefore, for inquiries $q_2$ and $q_3$, the expected response is "unsolvable" (as shown in Table 6), and when the LLM's response differs from the original answer, we consider it to have passed the corresponding inquiry. For the remaining seven dimensions and the original dataset, *Pass* refers to correctly answer.

### 4.2. Main Results

Table 1 presents the original results ("Vanilla") of all LLMs as well as their performance under our CogMath framework. First, there is a significant decrease of 30%-40% in pass rates for all models, indicating that the mathematical abili-

| | | MATH | | | | | | | | GSM8K | MExam |
|---|---|---|---|---|---|---|---|---|---|---|---|
| | | Avg | Alg | Count | Geo | Itmd | Num | Pre-Alg | Pre-Cal | | |
| GPT-4 | Stage 1 | 0.630 | 0.813 | 0.671 | 0.459 | 0.401 | 0.635 | 0.798 | 0.452 | 0.851 | 0.690 |
| | Stage 2 | **0.532** | **0.683** | **0.534** | **0.395** | **0.323** | **0.485** | **0.728** | **0.401** | 0.870 | 0.624 |
| | Stage 3 | 0.699 | 0.773 | 0.698 | 0.595 | 0.604 | 0.711 | 0.790 | 0.630 | **0.832** | **0.600** |
| GPT-3.5 | Stage 1 | 0.359 | 0.561 | 0.283 | 0.246 | **0.147** | 0.257 | 0.571 | **0.194** | **0.636** | 0.443 |
| | Stage 2 | **0.334** | **0.482** | **0.262** | **0.228** | 0.161 | **0.250** | **0.543** | 0.209 | 0.707 | **0.407** |
| | Stage 3 | 0.486 | 0.574 | 0.460 | 0.397 | 0.396 | 0.465 | 0.563 | 0.443 | 0.662 | 0.474 |
| Gemini-1.5 | Stage 1 | 0.509 | 0.715 | 0.428 | 0.388 | 0.307 | 0.415 | 0.692 | 0.372 | 0.829 | 0.618 |
| | Stage 2 | **0.421** | **0.586** | **0.380** | **0.284** | **0.240** | **0.300** | **0.629** | **0.302** | 0.806 | **0.579** |
| | Stage 3 | 0.659 | 0.741 | 0.660 | 0.534 | 0.571 | 0.678 | 0.718 | 0.623 | **0.748** | 0.653 |
| Llama3-8B | Stage 1 | 0.168 | 0.256 | 0.133 | 0.094 | **0.059** | **0.094** | 0.318 | **0.090** | 0.607 | 0.301 |
| | Stage 2 | **0.160** | **0.215** | **0.118** | **0.079** | 0.079 | 0.106 | **0.307** | 0.104 | 0.626 | **0.294** |
| | Stage 3 | 0.303 | 0.356 | 0.314 | 0.240 | 0.235 | 0.244 | 0.392 | 0.267 | **0.556** | 0.348 |
| Llama2-13B | Stage 1 | **0.039** | 0.063 | 0.076 | **0.019** | **0.019** | **0.012** | 0.085 | **0.011** | 0.243 | **0.118** |
| | Stage 2 | 0.047 | **0.062** | **0.027** | 0.029 | 0.037 | 0.024 | **0.080** | 0.037 | 0.253 | 0.133 |
| | Stage 3 | 0.117 | 0.132 | 0.122 | 0.081 | 0.113 | 0.094 | 0.140 | 0.103 | **0.232** | 0.289 |
| Mixtral-8x7B | Stage 1 | **0.200** | **0.308** | **0.131** | **0.127** | **0.094** | **0.113** | **0.327** | **0.150** | **0.400** | 0.364 |
| | Stage 2 | 0.224 | 0.328 | 0.139 | 0.146 | 0.136 | 0.133 | 0.344 | 0.185 | 0.430 | **0.332** |
| | Stage 3 | 0.398 | 0.434 | 0.376 | 0.372 | 0.341 | 0.337 | 0.490 | 0.374 | 0.569 | 0.432 |
| DeepSeek-V2.5 | Stage 1 | 0.649 | 0.844 | 0.578 | 0.507 | 0.455 | 0.683 | 0.780 | 0.491 | 0.832 | 0.717 |
| | Stage 2 | **0.526** | **0.695** | **0.496** | **0.411** | **0.328** | **0.463** | **0.723** | **0.357** | 0.850 | 0.672 |
| | Stage 3 | 0.681 | 0.762 | 0.692 | 0.610 | 0.607 | 0.644 | 0.741 | 0.623 | **0.817** | **0.541** |

Table 2: Performance of different LLMs at each cognitive stage.

ties they display on public benchmarks may not be as genuine and reliable as they appear. Even GPT-4 successfully passes only 39.3% and 67.1% of problems in MATH and GSM8K datasets, respectively. Second, on the more challenging MATH dataset, the most powerful models (i.e., perform best in "Vanilla"), GPT-4 and DeepSeek-V2.5, exhibit the largest drops, with $\Delta$ =36.5% and 37.9%, respectively. However, on the simpler GSM8K dataset, their declines are the smallest, with $\Delta$ =28.3% and 30.5%, respectively. This suggests that the extent to which the capabilities of LLMs are overestimated does not diminish as the models become stronger, but rather remains a widespread phenomenon unrelated to model size or dataset difficulty. Third, the issue of overestimated model capability persists on our newly constructed MExam dataset, which has not been used for training these LLMs. On one hand, this suggests that the overestimation is not solely due to data contamination. On the other hand, this phenomenon demonstrates that simply introducing more test problems may be insufficient to assess the true mathematical abilities of LLMs.

### 4.3. Analysis on Three Cognitive Stages

To further analyze the extent to which LLMs grasp different cognitive stages, we present the *Pass Rate* at different stages in Table 2, with the stage having the lowest pass rate highlighted in bold. Specifically, we first observe that for weaker LLMs (e.g., Llama2-13B), their pass rates in Stage 1 (i.e., *problem comprehension*) are the lowest, indicating that these models already exhibit deficiencies in fundamental understanding. For more advanced models (e.g., GPT-4,

DeepSeek-V2.5), their comprehension abilities appear more stable. However, they struggle significantly with mastering Stage 2 (e.g., GPT-4 and Deepseek exhibit pass rates of only 53.2% and 52.6% on MATH, respectively). In Section 4.4, we reveal that the main reason is that their grasp of knowledge is still unstable. Finally, the pass rate in Stage 3 remains below 0.85. This suggests that current LLMs are more suited for forward reasoning, i.e., generating answers based on the problems, but struggle to assess whether the solution aligns with the original problem from a backward perspective. This finding is consistent with existing research that shows LLMs may find it challenging to verify the correctness of their own answers (Huang et al., 2024).

### 4.4. Analysis on Nine Cognitive Dimensions

Furthermore, we analyze the performance of LLMs in each dimension. Specifically, for dimension $i$, we calculate a *Relative Pass Rate* (RPR) defined as: RPR = $\frac{|Pass_i \cap Pass|}{|Pass|}$. Here, $Pass_i$ denotes the problems where the LLM successfully passes their corresponding inquiry $q_i$, and $Pass$ refers to the problems correctly answered by this model. It is important to note that a higher RPR indicates better robustness and stability of the LLM's capabilities in that dimension. This is because the model's performance on corresponding inquiries is highly consistent with its performance on the original problems, making it less likely to exhibit defects when it answers the original problem correctly. Conversely, a lower RPR signifies a more detrimental impact on LLM performance, suggesting that the model exhibits lower adaptability to that type of inquiry.

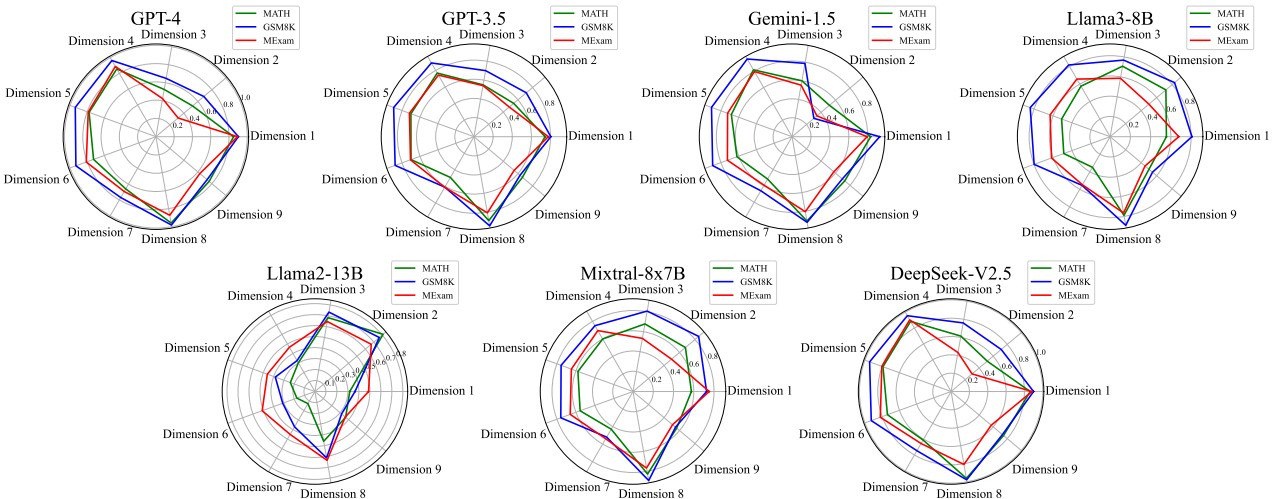

Figure 2: *Relative Pass Rate* (RPR) of different LLMs in each dimension.

Overall, from Figure 2, DeepSeek-V2.5 and GPT-4 exhibit the most balanced performance across multiple dimensions, followed by GPT-3.5, Mixtral-8x7B, Gemini-1.5, and Llama3-8B, with Llama2-13B performing the worst. Secondly, regarding the four dimensions in *problem comprehension* stage, an important observation is that GPT-4, GPT-3.5, Gemini-1.5, and DeepSeek-V2.5 underperform in Dimensions 2 and 3, even lagging behind Llama2-13B and Llama3-8B. We speculate that this is because most training data for current LLMs is composed of solvable math problems. After being trained on such data, when facing an unsolvable problem, current LLMs may inherently "over-correct" the problem into a solvable one, aligning it more closely with their training data. This insight suggests that in order to equip LLMs with more human-like cognitive capabilities, it is necessary to cultivate critical thinking skills rather than mere imitation of training data. Thirdly, for the three dimensions associated with *problem solving* stage, Dimension 7 accounts for the low pass rate discussed in Section 4.3. This indicates that current LLMs treat knowledge more as rigid memorization and application, rather than integrating it organically and flexibly into the reasoning process. Lastly, in *solution summarization* stage, nearly all LLMs demonstrate higher RPR values in Dimension 8, suggesting that they are quite adept at explaining reasoning steps. However, the performance in Dimension 9 indicates that these models struggle to use conclusions to reversely derive conditions, which explain why they are difficult to self-verify the correctness of their own answers.

## 4.5. Effect of LLM Enhancement Methods

We explore the impact of two commonly used reasoning enhancement methods on LLMs' mathematical abilities: Chain-of-Thought (CoT) (Wei et al., 2022) and In-Context

|       |           | CogMath | CogMath(CoT) | CogMath(ICL) |
|-------|-----------|---------|--------------|--------------|
|       | GPT-4     | 0.393   | 0.380        | 0.368        |
| MATH  | GPT-3.5   | 0.176   | 0.169        | 0.167        |
|       | Gemini-1.5| 0.291   | 0.242        | 0.250        |
|       | GPT-4     | 0.671   | 0.680        | 0.676        |
| GSM8K | GPT-3.5   | 0.424   | 0.442        | 0.466        |
|       | Gemini-1.5| 0.500   | 0.585        | 0.518        |

Table 3: Performances of LLM enhancement methods.

Learning (ICL) (Dong et al., 2022). For CoT, we prompt the LLM to answer each inquiry in CogMath "step by step". For ICL, we adopt a one-shot setting where, for each dimension $i$, we randomly sample a problem $P_i$ from the training set and use CogMath to construct an (inquiry $q_P^i$, answer $a_P^i$) pair as the demonstration.

From Table 3, these techniques led to a performance decrease of 0.7% ($0.176 \rightarrow 0.169$) to 4.9% ($0.291 \rightarrow 0.242$) on MATH but an increase of 0.5% ($0.671 \rightarrow 0.676$) to 8.5% ($0.500 \rightarrow 0.585$) on GSM8K. These results suggest that prompting techniques may not fundamentally enhance the mathematical abilities of LLMs. Instead, they serve more as an auxiliary tool, which can bring more positive effects on simpler datasets. In some cases, ICL might even limit the model's problem-solving flexibility. For more analyses of the effects at the dimension level, please refer to Appendix E.

## 4.6. Error Analysis

From Sections 4.2 to 4.4, we verify that the primary reason for LLMs making errors in our CogMath is due to their deficiencies in abilities corresponding to Dimensions 2, 3, 7, and 9. In this section, we further investigate how the characteristics of the problems influence LLMs' errors. Specifically, we take the MATH dataset as an example and explore the influence of problem difficulty and problem length. Prob-

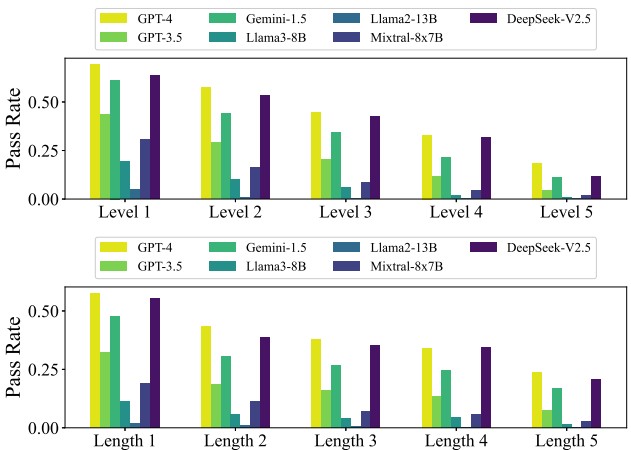

Figure 3: Relationship between LLM performance with problem characteristics (i.e., difficulty and length).

lem difficulty is measured by the dataset's inherent "level" labels, which include five tiers. For problem length, we divide all problems into five levels using an equal-frequency binning approach.

From Figure 3, we can first observe that as problem difficulty increases, the performance of all LLMs declines significantly. More specifically, most models only perform well on level 1 problems, while only GPT-4 and DeepSeek-V2.5 demonstrate proficiency on more than half of the problems at both levels 1 and 2. Secondly, as problem length increases, the LLM performance also shows some decline, though it is less significant compared to the impact of problem difficulty. This suggests that problem length has a relatively lower correlation with model performance. Based on these observations, we think future improvements in LLMs' mathematical abilities could focus on enhancing their capacity to handle more complex problems, particularly those in higher difficulty levels.

### 4.7. Human Verification of Agents

In this section, we provide evidence to make sure all our agents faithfully finish their jobs. Specifically, for each *Inquiry* agent, we have designed a *Judge* agent to evaluate the quality of its output. If the *Judge* agent determines that the output does not meet the required quality standards, we ask the *Inquiry* agent to regenerate the response.

To assess the effectiveness of the *Judge* agent, we invite 5 well-trained annotators with undergraduate degrees to verify the final inquires approved by the *Judge* agent, evaluating the rate at which they align with the intended dimensions. The evaluation template is same with the prompts for the *Judge* agent as presented in Appendix B. The evaluation protocol was approved by the Ethics Review Board and all annotators have been informed about the intended use of the data. The results for each dimension on 500 randomly

| Dimension | D.1 | D.3 | D.4 | D.5 | D.6 | D.7 | D.8 | D.9 |
|---|---|---|---|---|---|---|---|---|
| Rate (*Judge*) | 0.984 | 0.992 | 0.964 | 0.986 | 0.986 | 0.952 | 0.990 | 0.950 |

Table 4: Human Verification of *Judge* agents.

| Dimension | D.5 | D.6 | D.7 | D.8 |
|---|---|---|---|---|
| Rate (*Reference*) | 0.954 | 0.968 | 0.952 | 0.986 |

Table 5: Human Verification of *Reference* agents.

selected problems are shown in Table 4 (Since Dimension 2 relies on rule-based sentence disruption, there is no need for a *Judge* agent).

Regarding the *Reference* agents, Dimensions 1–4 simply use the original problem's answer as the correct response for $q_i$, while Dimension 9 automatically extracts the masked value from the *Inquiry* agent's output as $a_9$. Here, we focus on evaluating Dimensions 5–8. We invite the same annotators to evaluate the results of another 500 randomly selected problems. As shown in Table 5, the answers generated by the *Reference* agent achieve a pass rate of 95% across all dimensions, which ensures the quality of its outputs.

## 5. Conclusion and Discussion

In this paper, we introduced CogMath, a comprehensive and scientific evaluation framework that assesses the mathematical abilities of LLMs across three cognitive stages and nine dimensions of humans. The findings indicated that the abilities of current mainstream LLMs are overestimated by approximately 30%-40%. Moreover, we located the strength and weakness of different LLMs and verified that prompting techniques such as CoT and ICL do not genuinely enhance their mathematical proficiency. For future work, we discuss some valuable directions in Appendix F.

## Impact Statement

CogMath presents a valuable evaluation framework for assessing the reasoning capabilities of LLMs, but certain aspects warrant further consideration. Firstly, CogMath relies heavily on interactions among multiple LLM agents, which may limit the scalability due to the computational costs associated with generating inquiries for large-scale benchmarks. Secondly, while our study contributes to the understanding of LLM enhancement methods, it does not encompass all existing techniques, notably excluding widely adopted approaches such as Program of Thoughts (PoT) and Retrieval-Augmented Generation (RAG). These methods involve additional processes, such as code generation and information retrieval, which were beyond the scope of this work. Investigating whether these techniques can fundamentally enhance the reasoning abilities of LLMs remains an important direction for future research.

## Acknowledgement

This research was partially supported by the National Natural Science Foundation of China (Grants No.62477044, 62406303, U23A20319), Anhui Provincial Natural Science Foundation (No. 2308085QF229), the Fundamental Research Funds for the Central Universities (No.WK2150110038). Zhenya Huang gratefully acknowledges the support of the Young Elite Scientists Sponsorship Program by CAST (No. 2024QNRC001).

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

## A. An Example of CogMath

We present in Table 6 the inquiries across nine dimensions in CogMath for mathematical problem "Ali had \$21. Leila gave him half of her \$100. How much does Ali have now?".

## B. Prompts in CogMath Framework

The prompts for all agents across the 9 dimensions are presented in Figures B.1.1 to B.8.2. Notably, in CogMath, the expected answers for Dimensions 1 to 4 are the original answers $A$ of problem $P$, so we omit the corresponding *Reference* agents for these dimensions. For Dimension 2, the *Inquiry* agent automatically disrupts the word order in each clause according to rules, and this process does not require a special prompt or a *Judge* agent for evaluation. Hence, all agents for Dimension 2 are omitted here. As for Dimension 9, as shown in Figure A.8.1, its *Inquiry* agent also automatically determines the answer for inquiry $q_9$ (marked with "[]"), so there is no need to design an additional *Reference* agent prompt, which is therefore omitted.

## C. Implementation Details

All the *Inquiry* agents, *Reference* agents, and *Judge* agents are implemented with GPT-4. Besides, the maximum number of iterations for *Inquiry* agent is set to $\delta = 10$. If after 10 iterations, we still fail to obtain a satisfactory inquiry, we consider the problem to be unsuitable to be evaluated from that dimension. For such problems, we omit consideration of that dimension during the evaluation.

We apply CogMath on two of the most representative mathematical benchmarks, GSM8K (Cobbe et al., 2021) and MATH (Hendrycks et al., 2021), along with our constructed MExam dataset. GSM8K is an elementary-level math word problem dataset that primarily involves basic understanding and reasoning. MATH is a high school competition-level dataset, consisting of 7 subcategories, such as algebra and geometry. MExam is composed of 6,353 questions manually collected from real exams in 50 Chinese exercise books, which covers the full K-12 mathematics curriculum. For GSM8K and MATH, since their training sets may have already been used in the training process of current LLMs, we apply CogMath on their public test sets, which contain 1,319 and 5,000 questions, respectively.

## D. Evaluation of DeepSeek-R1

Due to the cost and rate limitations of API calls, here we evaluate the current SOTA DeepSeek-R1 (Guo et al., 2025) on the most widely used GSM8K and MATH datasets in Table 7. First, compared to Table 1, DeepSeek-R1 achieves the best performance among all evaluated LLMs, both in "Vanilla" and CogMath framework. This shows its supe-

| Stages | Dimensions | Example of Inquiry $q_i$ | Pass |
|---|---|---|---|
| Problem Comprehension | **Dimension 1:** Sentence Paraphrasing | Jacob had $21. Emily shared half of her $100 with him. How much money does Jacob have now? | Answer Correctly |
| | **Dimension 2:** Sentence Disruption | $21 Ali had half of $100 him Leila her gave now? does Ali much How have | Identify "Unsolvable" |
| | **Dimension 3:** Missing Condition | Ali had some money. Leila gave him half of her money. How much does Ali have now? | Identify "Unsolvable" |
| | **Dimension 4:** Redundant Condition | Ali had $21. Leila gave him half of her $100. Before meeting with Leila, Ali had already counted his money twice to make sure it was correct. How much does Ali have now? | Answer Correctly |
| Problem Solving | **Dimension 5:** Analogical Reasoning | Tom had $21 comic books. Jerry traded him half of his collection of $100 comic books. How many comic books does Tom have now? | Answer Correctly |
| | **Dimension 6:** Numerical Transformation | Ali had $30. Leila gave him half of her $120. How much does Ali have now? | Answer Correctly |
| | **Dimension 7:** Knowledge Redefinition | Assume "half" means one-third of the given amount, solve the following problem: Ali had $21. Leila gave him half of her $100. How much does Ali have now? | Answer Correctly |
| Solution Summarization | **Dimension 8:** Intermediate Step Questioning | Given the mathematical problem: Ali had $21. Leila gave him half of her $100. How much does Ali have now? please answer my following question: Why does Ali now have $71? | Answer Correctly |
| | **Dimension 9:** Backward Reasoning | In the problem, "Ali had $21. Leila gave him half of her $\alpha$, where $\alpha$ is an unknown total amount of money Leila had. How much does Ali have now?", if Ali now has $71, what is the value of $\alpha$? | Answer Correctly |

Table 6: The 3 cognitive stages and 9 dimensions in our CogMath. "Pass" refers to the type of LLM response that is considered to pass the inquiry $q_i$ of the given dimension.

rior mathematical reasoning capabilities. Second, based on the performance gap (marked as $\Delta$), DeepSeek-R1 still exhibits a certain degree of overestimation, highlighting the necessity of our proposed evaluation from the human cognitive perspective. Third, similar to other advanced LLMs, DeepSeek-R1 encounters the most challenges in Stage 2 (i.e., *Problem Solving*). Further analysis reveals that its main weakness lies in Dimension 7 (Knowledge Redefinition), with a *Relative Pass Rate* (RPR) of 0.617. This supports the conclusion that current LLMs rely on fixed memorization rather than adapting knowledge flexibly. Fourth, compared to DeepSeek-V2.5, DeepSeek-R1 improves significantly in Stage 3 (*Solution Summarization*), suggesting a deeper understanding of reasoning process.

## E. Dimension-level Effects of LLM enhancement methods

Here we present the effects of ICL on each dimension. As defined in Section 4.5, the higher *Pass Rate* represents better performance.

From Table 8, we observe that Dimensions 6 and 7 show consistently stable improvements with the introduction of ICL for all LLMs. This suggests that ICL has a significant positive impact on the reasoning abilities in handling numerical transformations and knowledge redefinition. Second, Dimension 5 benefits the least from ICL, and it consistently experiences negative effects. This may be due to the reason-

ing process in the demonstration diverging from that of the original problem, which could disrupt the model's performance in analogical reasoning (i.e., reasoning that follows the same process as the original problem). Third, for other dimensions, the effect of ICL varies across different models. For instance, in Dimension 3, GPT-4 and GPT-3.5 show improvements, while Gemini-1.5 shows a decline. This highlights the differing robustness and capabilities of various models, providing insights into potential weaknesses and future development directions of different LLMs.

## F. Discussion

First, our CogMath framework is highly generalizable, as it does not rely on specific problem types or formats, making it applicable to testing LLMs' cognitive abilities in other mathematical tasks, such as theorem proving. Besides, beyond evaluating individually, we can also compose multiple dimensions for assessing human-like multi-behaviors (Zhang et al., 2025b) and deeper cognitive diagnosis (Liu et al., 2021). Second, our framework can be easily extended to tasks in other domains. For instance, in visual reasoning tasks, a *visual comprehension* stage could be added into our framework, along with dimensions like image perturbation to evaluate the capabilities and robustness of visual LLMs like GPT-4v. Third, through experiments in Sections 4.2 to 4.6, we have conducted a detailed examination of LLMs' mastery across different dimensions, providing valuable insights for future model improvements. For example, as

| | | MATH | | | | | | | | GSM8K |
|---|---|---|---|---|---|---|---|---|---|---|
| | | **Avg** | **Alg** | **Count** | **Geo** | **Itmd** | **Num** | **Pre-Alg** | **Pre-Cal** | |
| DeepSeek-R1 | Vanilla | 0.982 | 0.992 | 0.994 | 0.956 | 0.979 | 0.980 | 0.985 | 0.972 | 0.967 |
| | CogMath | 0.448 | 0.581 | 0.443 | 0.307 | 0.295 | 0.413 | 0.604 | 0.326 | 0.703 |
| | Δ | -0.534 | -0.411 | -0.551 | -0.649 | -0.684 | -0.567 | -0.381 | -0.646 | -0.264 |
| | *Stage* 1 | 0.863 | 0.942 | 0.831 | 0.737 | 0.837 | 0.881 | 0.875 | 0.837 | 0.897 |
| | *Stage* 2 | **0.575** | **0.715** | **0.557** | **0.441** | **0.405** | **0.544** | **0.738** | **0.456** | 0.848 |
| | *Stage* 3 | 0.753 | 0.808 | 0.773 | 0.637 | 0.694 | 0.743 | 0.815 | 0.725 | **0.856** |

Table 7: Performance of DeepSeek-R1.

| MATH | | D.1 | D.2 | D.3 | D.4 | D.5 | D.6 | D.7 | D.8 | D.9 |
|---|---|---|---|---|---|---|---|---|---|---|
| GPT-4 | CogMath | 0.737 | 0.589 | 0.596 | 0.732 | 0.695 | 0.632 | 0.611 | 0.943 | 0.727 |
| | CogMath(ICL) | 0.722 | 0.557 | 0.645 | 0.714 | 0.638 | 0.681 | 0.625 | 0.930 | 0.720 |
| | Δ | -1.5% | -3.2% | -4.9% | -1.8% | -5.7% | +4.9% | +1.4% | -1.3% | -0.7% |
| GPT-3.5 | CogMath | 0.486 | 0.666 | 0.700 | 0.486 | 0.509 | 0.457 | 0.379 | 0.823 | 0.556 |
| | CogMath(ICL) | 0.504 | 0.582 | 0.733 | 0.498 | 0.494 | 0.499 | 0.434 | 0.832 | 0.559 |
| | Δ | +1.8% | -8.4% | +3.3% | +1.2% | -1.5% | +4.2% | +5.5% | +0.9% | +0.3% |
| Gemini-1.5 | CogMath | 0.602 | 0.569 | 0.764 | 0.591 | 0.603 | 0.531 | 0.467 | 0.887 | 0.696 |
| | CogMath(ICL) | 0.595 | 0.662 | 0.730 | 0.589 | 0.543 | 0.545 | 0.466 | 0.870 | 0.619 |
| | Δ | -0.7% | +9.3% | -3.4% | -0.2% | -6.0% | +1.4% | -0.1% | -1.7% | -7.7% |
| GSM8K | | D.1 | D.2 | D.3 | D.4 | D.5 | D.6 | D.7 | D.8 | D.9 |
| GPT-4 | CogMath | 0.886 | 0.692 | 0.657 | 0.946 | 0.930 | 0.921 | 0.792 | 0.976 | 0.828 |
| | CogMath(ICL) | 0.889 | 0.662 | 0.754 | 0.943 | 0.920 | 0.928 | 0.801 | 0.972 | 0.853 |
| | Δ | +0.3% | -3.0% | +9.7% | -0.3% | -1.0% | +0.7% | +0.9% | -0.4% | +2.5% |
| GPT-3.5 | CogMath | 0.730 | 0.741 | 0.728 | 0.816 | 0.833 | 0.792 | 0.589 | 0.899 | 0.668 |
| | CogMath(ICL) | 0.778 | 0.640 | 0.773 | 0.802 | 0.810 | 0.826 | 0.592 | 0.901 | 0.704 |
| | Δ | +4.8% | -10.1% | +4.5% | -1.4% | -2.3% | +3.4% | +0.3% | +0.2% | +3.6% |
| Gemini-1.5 | CogMath | 0.773 | 0.730 | 0.985 | 0.821 | 0.890 | 0.873 | 0.672 | 0.907 | 0.786 |
| | CogMath(ICL) | 0.807 | 0.763 | 0.859 | 0.895 | 0.873 | 0.868 | 0.697 | 0.861 | 0.773 |
| | Δ | +3.4% | +3.3% | -12.6% | +7.4% | -1.7% | -0.5% | +2.5% | -4.6% | -1.3% |

Table 8: Dimension-level Performances of In-Context Learning.

observed in Section 4.4, existing LLMs may exhibit an "over-correction" behavior when faced with unsolvable problems. To address this, we need to introduce critical thinking mechanisms that enable them to reconsider the fundamental nature of each problem (Zhang et al., 2025a), rather than merely imitating patterns from training data. Lastly, from the results of Section 4.5, we found that CoT and ICL may not fundamentally improve the mathematical capabilities of LLMs. However, these techniques have been shown to enhance performance in many NLP tasks. Therefore, we believe that understanding the underlying mechanisms of these methods from a theoretical perspective remains a critical research question.

> **B.1.1: Dimension 1 (*Inquiry* agent) prompt**
>
> Now you are a question rewriting agent with interleaving Thought, Action. Thought can be the process of rewriting. Action MUST BE THE REWRITE QUESTION WHICH REPHRASES THE QUESTION. You will be provided with a math problem. Please rephrase the question in a different way.
> Here are an example:
> Question: Gloria is shoe shopping when she comes across a pair of boots that fit her shoe budget. However, she has to choose between the boots and two pairs of high heels that together cost five dollars less than the boots. If one pair of heels costs $33 and the other costs twice as much, how many dollars are the boots.
> Thought: Identify the main elements (character Gloria, items boots and high heels, price relationship), analyze the price relationship (high heels cost $33 and $66, total $99 is $5 less than boots, making boots $104), find the logical relationship (choice, price relationship, and calculation), and change the character (to Alice) and item names (to sneakers and sandals) to rewrite the problem while keeping the prices and relationships the same.
> Action: Alice is shopping for footwear when she finds a pair of sneakers that fit her budget. However, she has to choose between the sneakers and two pairs of sandals that together cost five dollars less than the sneakers. If one pair of sandals costs $33 and the other costs twice as much, how many dollars are the sneakers?
>
> Now, here is your question:
> Question: {Here is the original problem P}

**B.1.2: Dimension 1 (*Judge* agent) prompt**

Now you are a judge agent with interleaving Thought, Action. Your task is to determine if the rewritten question is a rephrased version of the original question. Thought can be articulating the logical relationship between the original question and the rewritten question, and analyze whether the logical relationship between the two is consistent. Action must be Yes or No.

Here are an example:

Original question: Janet's ducks lay 16 eggs per day. She eats three for breakfast every morning and bakes muffins for her friends every day with four. She sells the remainder at the farmers' market daily for $2 per fresh duck egg. How much in dollars does she make every day at the farmers' market?

Rewritten question: Kevin's chickens lay 16 eggs each day. He consumes three for his morning meal and uses four to make pastries each day for his neighbors. The remaining eggs are sold at the local farmers' market daily for $2 per egg. How much money does Kevin earn from egg sales each day at the market?

Thought:

Original Question Key Elements:

Subject: Janet's ducks

Daily egg production: 16 eggs

Daily consumption: 3 eggs for breakfast, 4 eggs for muffins

Selling price: $2 per egg

Question focus: Daily earnings from selling eggs at the farmers' market

Rewritten Question Key Elements:

Subject: Kevin's chickens

Daily egg production: 16 eggs

Daily consumption: 3 eggs for morning meal, 4 eggs for pastries

Selling price: $2 per egg

Question focus: Daily earnings from selling eggs at the farmers' market

the rewritten question is a rephrased version of the original question. Both questions convey the same information and ask the same type of question, with only the subject (Janet's ducks vs. Kevin's chickens) and the specific uses of the eggs (breakfast vs. morning meal, muffins vs. pastries) being slightly different. The logical relationship between the two questions is consistent.

Action:Yes

Now, here are your raw question and rewritten question:

Original question:{Here is the original problem P}

Rewritten question:{Here is the inquiry $q_1$}

---

**B.2.1: Dimension 3 (*Inquiry* agent) prompt**

Now you are a question rewriting agent with interleaving Thought, Action. Thought can reason about the necessary conditions for the question. Action MUST BE THE REWRITE QUESTION WHICH REMOVE THE NECESSARY CONDITION.

You will be provided with a math problem. Please analyze the necessary conditions, remove one necessary condition, and make the problem unsolvable.

Here are an example:

Question: Gloria is shoe shopping when she comes across a pair of boots that fit her shoe budget. However, she has to choose between the boots and two pairs of high heels that together cost five dollars less than the boots. If one pair of heels costs $33 and the other costs twice as much, how many dollars are the boots.

Thought: Necessary conditions are

1.Gloria has to choose between purchasing a pair of boots or two pairs of high heels.

2.The price of one pair of high heels is $33.

3.The price of the other pair of high heels is twice the price of the first pair, which is $66.

Action: Gloria is shoe shopping when she comes across a pair of boots that fit her shoe budget. However, she has to choose between the boots and two pairs of high heels. If one pair of heels costs $33 and the other costs twice as much, how many dollars are the boots?

Now, here is your question:

Question:{Here is the original problem P}

---

**B.2.2: Dimension 3 (*Judge* agent) prompt**

Now you are a judge agent with interleaving Thought, Action. Your task is to determine if the rewritten question lacks a crucial condition compared to the original question. Thought can be the comparison of the key conditions of two questions. Action must be Yes or No.

Here are an example:

Original question: Janet's ducks lay 16 eggs per day. She eats three for breakfast every morning and bakes muffins for her friends every day with four. She sells the remainder at the farmers' market daily for $2 per fresh duck egg. How much in dollars does she make every day at the farmers' market?

Rewritten question: Janet's ducks lay 16 eggs per day. She eats three for breakfast every morning and bakes muffins for her friends every day with four. She sells the remainder at the farmers' market daily. How much in dollars does she make every day at the farmers' market?

Thought: Original Question Key Conditions:

Janet's ducks lay 16 eggs per day.

She eats three eggs for breakfast every morning.

She bakes muffins with four eggs every day.

She sells the remainder at the farmers' market daily for $2 per fresh duck egg.

Rewritten Question Key Conditions:

Janet's ducks lay 16 eggs per day.

She eats three eggs for breakfast every morning.

She bakes muffins with four eggs every day.

She sells the remainder at the farmers' market daily.

The rewritten question lacks the crucial condition specifying the price per egg sold at the farmers' market ($2 per fresh duck egg), which is essential for calculating the daily earnings.

Action: Yes

Now, here are your raw question and rewritten question:

Original question:{Here is the original problem P}

Rewritten question:{Here is the inquiry $q_3$}

---

**B.3.1: Dimension 4 (*Inquiry* agent) prompt**

Now you are a question rewriting agent with interleaving Thought, Action. Thought can reason about the unnecessary condition for the question. Action MUST BE THE REWRITE QUESTION WHICH ADDS A UNNECESSARY CONDITION.

You will be provided with a math problem. Please analyze the necessary conditions, add one unnecessary condition, and make the problem solvable.

Here are an example:

Question: Gloria is shoe shopping when she comes across a pair of boots that fit her shoe budget. However, she has to choose between the boots and two pairs of high heels that together cost five dollars less than the boots. If one pair of heels costs $33 and the other costs twice as much, how many dollars are the boots.

Thought: The added irrelevant condition: Gloria's friend informed her that the boots are on sale for a 10% discount, but this information is incorrect and does not affect the actual price of the boots.

Action: Gloria is shoe shopping when she comes across a pair of boots that fit her shoe budget. However, she has to choose between the boots and two pairs of high heels that together cost five dollars less than the boots. If one pair of heels costs $33 and the other costs twice as much, how many dollars are the boots. Additionally, Gloria's friend told her that the boots are on sale for 10% off, but this information is incorrect and does not affect the actual price of the boots.

Now, here is your question:

Question:{Here is the original problem P}

### B.3.2: Dimension 4 (*Judge* agent) prompt

Now you are a judge agent with interleaving Thought, Action. Your task is to determine if the rewritten question has an additional condition compared to the original question. Thought can be the comparison of the key conditions of two questions. Action must be Yes or No.
Here are an example:
Original question: Janet's ducks lay 16 eggs per day. She eats three for breakfast every morning and bakes muffins for her friends every day with four. She sells the remainder at the farmers' market daily for $2 per fresh duck egg. How much in dollars does she make every day at the farmers' market?
Rewritten question: Janet's ducks lay 16 eggs per day. She eats three for breakfast every morning and bakes muffins for her friends every day with four. She sells the remainder at the farmers' market daily for $2 per fresh duck egg. Additionally, she uses exactly two eggs every Sunday to make a special omelette for her family, but this does not affect her daily revenue. How much in dollars does she make every day at the farmers' market?"
Thought:
Original Question Key Conditions:
Janet's ducks lay 16 eggs per day.
She eats three eggs for breakfast every morning.
She bakes muffins with four eggs every day.
She sells the remainder at the farmers' market daily for $2 per fresh duck egg.

Rewritten Question Key Conditions:
Janet's ducks lay 16 eggs per day.
She eats three eggs for breakfast every morning.
She bakes muffins with four eggs every day.
She sells the remainder at the farmers' market daily for $2 per fresh duck egg.
Additionally, she uses exactly two eggs every Sunday to make a special omelette for her family, but this does not affect her daily revenue.

the rewritten question has an additional condition regarding the use of two eggs every Sunday for a special omelette, which is not present in the original question.
Action: Yes

Now, here are your raw question and rewritten question:
Original question:{Here is the original problem P}
Rewritten question:{Here is the inquiry $q_4$}

### B.4.1: Dimension 5 (*Inquiry* agent) prompt

Now you are a question rewriting agent. Please modify the context of the question to test a student's ability to apply their knowledge in different scenarios. While modifying the context, you must not change the solution approach or the specific numerical values in the problem.

Here are an example:
Question: Janet's ducks lay 16 eggs per day. She eats three for breakfast every morning and bakes muffins for her friends every day with four. She sells the remainder at the farmers' market daily for $2 per fresh duck egg. How much in dollars does she make every day at the farmers' market?
Rewritten Question: David's apple trees produce 16 apples per day. He eats three for lunch every afternoon and uses four to make apple pies for his neighbors each day. He sells the remainder at the local grocery store daily for $2 per fresh apple. How much in dollars does he make every day at the grocery store?

Now, here is your question:
Question:{Here is the original problem P}

### B.4.2: Dimension 5 (*Judge* agent) prompt

Now you are a judge agent with interleaving Thought, Action. Your task is to determine if the rewritten question uses the same knowledge points of the original question and only changes the application scene. Thought can be articulating the logical relationship between the original question and the rewritten question, and analyze their knowledge points and application scene. Action must be Yes or No.
Here are an example:
Original question: Janet's ducks lay 16 eggs per day. She eats three for breakfast every morning and bakes muffins for her friends every day with four. She sells the remainder at the farmers' market daily for $2 per fresh duck egg. How much in dollars does she make every day at the farmers' market?
Rewritten question: David's apple trees produce 16 apples per day. He eats three for lunch every afternoon and uses four to make apple pies for his neighbors each day. He sells the remainder at the local grocery store daily for $2 per fresh apple. How much in dollars does he make every day at the grocery store?
Thought:
The solutions of the two question are similar, and both of them only use the basic knowledge of addition and subtraction and income formula. The original question is the application scene where Janet sells duck eggs. The rewritten question is the application scene where David sells apples, so their scenes are different.
Action: Yes

Now, here are your raw question and rewritten question:
Original question:{Here is the original problem P}
Rewritten question:{Here is the inquiry $q_5$}

### B.4.3: Dimension 5 (*Reference* agent) prompt

Now you are a solver agent with interleaving Thought, Action. Your task is to generate the New Answer for the New Question based on the Original Answer of the Original Question. Thought can be to refer to each step of the Original Answer.
Here are an example:
Original Question: Janet's ducks lay 16 eggs per day. She eats three for breakfast every morning and bakes muffins for her friends every day with four. She sells the remainder at the farmers' market daily for $2 per fresh duck egg. How much in dollars does she make every day at the farmers' market?
Original Answer: Janet sells 16 - 3 - 4 = <<16-3-4=9>>9 duck eggs a day.\nShe makes 9 * 2 = $<<9*2=18>>18 every day at the farmer\u2019s market.\n#### 18
New Question: Tom's lemon trees yield 16 lemons per day. He drinks juice made from three for breakfast every morning and uses four to prepare lemonade for his co-workers each day. He sells the remainder at the local outdoor market daily for $2 per fresh lemon. How much in dollars does he make every day at the market?
Thought:
The context changing from eggs to lemons and Janet to Tom.
Action: Tom lefts 16 - 3 - 4 = <<16-3-4=9>>9 lemons a day.\nHe makes 9 * 2 = $<<9*2=18>>18 every day at the local outdoor market.\n#### 18

Now, here are your raw question and rewritten question:
Original Question:{Here is the original problem P}
Original Answer:{Here is the original answer A}
New Question:{Here is the inquiry $q_5$}

### B.5.1: Dimension 6 (*Inquiry* agent) prompt

Now you are a question rewriting agent. Please change the numerical values in the problem, but during the modification, you must not alter the solution approach or the specific context of the problem. The modified values should still be consistent with the meaning of the original problem.
Here are an example:
Question: If a snack-size tin of peaches has $40$ calories and is $2\%$ of a person\'s daily caloric requirement, how many calories fulfill a person\'s daily caloric requirement?
Rewritten Question: If a snack-size tin of peaches has $60$ calories and is $3\%$ of a person\'s daily caloric requirement, how many calories fulfill a person\'s daily caloric requirement?

Now, here is your question:
Question:{Here is the original problem P}

### B.5.2: Dimension 6 (*Judge* agent) prompt

Now you are a judge agent with interleaving Thought, Action. Your task is to determine if the rewritten question only changes the numbers in the original question. Thought can be articulating the logical relationship between the original question and the rewritten question, and analyze whether their difference is only in numbers. Action must be Yes or No.
Here are two examples:
Original question: Three vertices of a cube in space have coordinates $A = (2,3,0),$ $B = (0,5,4),$ and $C = (4,1,8).$ Compute the coordinates of the center of the cube.
Rewritten question: Three vertices of a cube in space have coordinates \\(A = (3, 2, 1),\\ B = (1, 4, 5),\\) and \\(C = (5, 0, 9).\\ Compute the coordinates of the center of the cube.
Thought:
The difference between Original Question and Rewritten question is the coordinates of three points A,B and C. Thus, their difference is only in numbers.
Action: Yes

Original question: John adopts a dog. He takes the dog to the groomer, which costs $100. The groomer offers him a 30% discount for being a new customer. How much does the grooming cost?
Rewritten question: John adopts a cat. He takes the cat to the groomer, which costs $120. The groomer offers him a 25% discount for being a new customer. How much does the grooming cost?
Thought:
The Rewritten question not only changes the number $100 and 30%, but also change "dog" to "cat", which change the meaning of the Original question.
Action: No

Now, here are your raw question and rewritten question:
Original question:{Here is the original problem P}
Rewritten question:{Here is the inquiry $q_6$}

### B.5.3: Dimension 6 (*Reference* agent) prompt

You are a math expert. Please refer to the Original Answer of the Original Question to generate the answer of the New Question.
Original Question: {Here is the original problem P}
Original Answer: {Here is the original answer A}
New Question: {Here is the inquiry $q_6$}

### B.6.1: Dimension 7 (*Inquiry* agent) prompt

Now you are a question rewriting agent. Please redefine some mathematical concepts within the problem to test a student's learning outcomes. For a mathematical concept in the problem, you can change its definition. For example, you can redefine the formula for perimeter or area, but during the redefinition, do not change the original values or context of the problem.
Here are an example:
Question: You draw a rectangle that is 7 inches wide. It is 4 times as long as it is wide. What is the area of the rectangle?
Rewritten Question: Assume the area formula of a rectangle is the sum of its length and width, solve the following problem: You draw a rectangle that is 7 inches wide. It is 4 times as long as it is wide. What is the area of the rectangle?

Now, here is your question:
Question:{Here is the original problem P}

### B.6.2: Dimension 7 (*Judge* agent) prompt

Now you are a judge agent with interleaving Thought, Action. Your task is to determine if the two math problems use relatively close knowledge points (we allow differences in the definition or formula used in the solution process). Thought can involve articulating the logical relationship between the two math problems and analyzing their knowledge points. Action must be Yes or No.
Here are an example:
Original question: A sphere is inscribed inside a hemisphere of radius 2. What is the volume of this sphere?
Rewritten question: Assuming the volume of a sphere is calculated by twice the cube of the radius, rather than using the factor $\frac{4}{3}\pi$, solve the problem: A sphere is inscribed inside a hemisphere of radius 2. What is the volume of this sphere?
Thought:
Both problems deal with calculating the volume of a sphere, but the rewritten problem uses a modified formula for the volume (twice the cube of the radius instead of the standard $\frac{4}{3}\pi r^3$). While the specific formula is altered, the core knowledge point—understanding the volume of a sphere and its relationship to the radius—is the same.
Action: Yes

Now, here are your raw question and rewritten question:
Original question:{Here is the original problem P}
Rewritten question:{Here is the inquiry $q_7$}

### B.6.3: Dimension 7 (*Reference* agent) prompt

Please solve the following problem based on its assumption step by step: {Here is the inquiry $q_7$}

### B.7.1: Dimension 8 (*Inquiry* agent) prompt

Now you are a questioning agent with interleaving Thought, Action. Thought can choose one of the steps in the problem reasoning process. Action MUST BE A QUESTION ABOUT THE STEP.
You will be provided with a math problem and its reasoning process. Please choose a step, and ask a question about this step.

Here is an example:
Question: Gloria is shoe shopping when she comes across a pair of boots that fit her shoe budget. However, she has to choose between the boots and two pairs of high heels that together cost five dollars less than the boots. If one pair of heels costs $33 and the other costs twice as much, how many dollars are the boots.
Reasoning Process: The second pair of heels costs 33 * 2 = \$<<33*2=66>>66.\nThe heels together cost 66 + 33 = \$<<66+33=99>>99.\nThe boots cost \$5 more than both pairs of heels together, so the boots cost 99 + 5 = \$104.\n#### 104
Thought: The reasoning process consists of three steps, choose the second step that calculates the cost of heels.
Action: Why do the heels together cost 99.

Now, here is your question:
Question:{Here is the original problem P}

### B.7.2: Dimension 8 (*Judge* agent) prompt

Now you are a judge agent with interleaving Thought, Action. Your task is to determine if the given question is a correct question about the reasoning process of the problem. Thought can be the comparison between the question and the reasoning process of the problem. Action must be Yes or No.
Here are an example:
Original question: Janet's ducks lay 16 eggs per day. She eats three for breakfast every morning and bakes muffins for her friends every day with four. She sells the remainder at the farmers' market daily for $2 per fresh duck egg. How much in dollars does she make every day at the farmers' market?
Reasoning process: Janet sells 16 - 3 - 4 = <<16-3-4=9>>9 duck eggs a day.\nShe makes 9 * 2 = \$<<9*2=18>>18 every day at the farmer\u2019s market.\n#### 18.
Given question: Why does Janet sell exactly 9 duck eggs a day?
Thought:
Steps in Reasoning process:
Janet sells 9 duck eggs a day.
She makes 18 every day.

The given question asks why does Janet sell 9 duck eggs a day, which is coincident with the reasoning process because the first step explains that Janet sells 9 duck eggs a day.
Action: Yes

Now, here are your raw question and rewritten question:
Original question:{Here is the original problem P}
Reasoning process:{Here is the original answer A}
Given question:{Here is the inquiry $q_8$}

### B.7.3: Dimension 8 (*Reference* agent) prompt

You are a math expert. Please answer my question about the mathematical problem based on the solution: {Here is the original problem P}
Solution: {Here is the original answer A}
My question is: {Here is the inquiry $q_8$}

### B.8.1: Dimension 9 (*Inquiry* agent) prompt

Now you are a question generating agent with interleaving Thought, Action. Thought can choose one of the numeric value in the problem and express it explicitly with []. Action MUST BE A QUESTION THAT MASKS THE NUMERIC VALUE AND ASKS TO DERIVE THE NUMERIC VALUE.
You will be provided with a math problem and its reasoning process. Please choose a numeric value from the problem, mask it with an unknown Greek letter, and generate a question that asks to derive the numeric value.
Here are an example:
Question: Gloria is shoe shopping when she comes across a pair of boots that fit her shoe budget. However, she has to choose between the boots and two pairs of high heels that together cost five dollars less than the boots. If one pair of heels costs $33 and the other costs twice as much, how many dollars are the boots.
Reasoning Process: The second pair of heels costs 33 * 2 = \$<<33*2=66>>66.\nThe heels together cost 66 + 33 = \$<<66+33=99>>99.\nThe boots cost $5 more than both pairs of heels together, so the boots cost 99 + 5 = $104.\n#### 104
Thought: Mask the number \"five\" in the sentence \"cost five dollars less than the boots\". The value of this number is [5].
Action: For problem \"Gloria is shoe shopping when she comes across a pair of boots that fit her shoe budget. However, she has to choose between the boots and two pairs of high heels that together cost x dollars less than the boots, where x is an unknown value. If one pair of heels costs $33 and the other costs twice as much, how many dollars are the boots.\", we know the answer for this problem is 104, find the value of x.

Now, here is your question:{Here is the original problem P}

### B.8.2: Dimension 9 (*Judge* agent) prompt

Now you are a judge agent with interleaving Thought, Action. Your task is to determine 1) whether the given question does not change the structure of the original question except that an unknown variable is introduced, 2) whether the given question is solvable, and 3) whether the answer to the question is new_answer. Thought can be the comparison between the question and the original question. Action must be Yes or No.
Here are two examples:
Original question: Janet's ducks lay 16 eggs per day. She eats three for breakfast every morning and bakes muffins for her friends every day with four. She sells the remainder at the farmers' market daily for $2 per fresh duck egg. How much in dollars does she make every day at the farmers' market?
Given question: Assuming Janet sells each duck egg at $\alpha$ dollars, where $\alpha$ is unknown. Given that she sells 9 eggs daily, and makes a total of 18 dollars from these sales, what is the value of $\alpha$ in dollars per egg?
New answer: 2
Thought:
The given question states that Janet sells 9 eggs daily, which is not mentioned in the original question. Therefore, the given question changes the semantics of the original question.
Action: No

Original question: A robe takes 2 bolts of blue fiber and half that much white fiber. How many bolts in total does it take?
Given question: For the problem \"A robe takes $\alpha$ bolts of blue fiber and half that much white fiber. How many bolts in total does it take?\", where $\alpha$ is an unknown value. If the total amount of bolts needed is 3, find the value of $\alpha$.
New answer: 2
Thought:
The given question has the same structure of the original question, with only replacing 2 with unknown valuable $\alpha$. Besides, the given question is solvable. Substitute $\alpha$ with 2, the total amount of bolts needed is still 3. Therefore, the answer to the given question is new answer.
Action: Yes

Now, here are your raw question and rewritten question:
Original question:{Here is the original problem P}
Given question:{Here is the inquiry $q_9$}
New answer: {Here is the new_answer $a_9$}

