# OpenReview forum: "CogMath: Assessing LLMs' Authentic Mathematical Ability from a Human Cognitive Perspective"
_ICML.cc/2025/Conference — ICML 2025 poster_

### Official Review · Reviewer_5FjB · 2025-03-10

**Overall Recommendation:** 4

**Summary:**

The paper introduces CogMath, an evaluation framework for assessing LLMs' capabilities from a human cognitive perspective. It breaks down mathematical reasoning into three stages: problem comprehension, problem solving, and solution summarization. Each stage is further evaluated through nine detailed dimensions, with three agents designed for scientific assessment per dimension. The experiments provide valuable insights into the strengths and weaknesses of seven mainstream LLMs, guiding future development.

**Claims And Evidence:**

The experiments on seven representative LLMs are significant, and the authors investigate their performances at both the stage and dimension levels. Besides, the authors also assess the impact of some LLM-enhanced methods including CoT and ICL, which greatly enriches the conclusion.

**Essential References Not Discussed:**

None.

**Experimental Designs Or Analyses:**

I have checked the details of the experimental process. The settings and metrics appear well-founded. Besides, I have read all the analyses and I think they are reasonable and insightful.

**Methods And Evaluation Criteria:**

The proposed three stages and nine dimensions in CogMath framework are grounded in psychological research on human reasoning process. Therefore, I believe they have great foundation. Besides, the authors provide human verification results for agents in each dimension. The results in Section 4.7 clearly demonstrate their quality and effectiveness.

**Other Comments Or Suggestions:**

Typos:
1. Line 48, “performance rom" should be “performance from”?
2. Line 60, should "Solution" be lowercase?
3. Line 67, “refine” should be “refines”

**Other Strengths And Weaknesses:**

The main strengths of this paper include the following:
1. The motivation to evaluate LLMs’ performances at human-level dimensions/stages is important and innovative. Besides, in each dimension, the authors introduce a judge agent and a reference agent to ensure the quality of the evaluation.
2. The authors conduct extensive experiments across several LLMs. The results provide detailed insights into the advantages and disadvantages of different LLMs. The exploration of LLM-enhanced methods such as CoT and ICL provides a good insight to prompt further study.
3. The paper is well-organized and easy to understand. The authors give sufficient examples to explain their ideas and implementations.

However, I have the following concerns and suggestions:
1. Some technical details in this paper require further clarification. Please refer to the questions below.
2. Since a key contribution of this paper is providing insights for improving LLMs, the authors should offer more detailed illustrations on this topic. Please refer to the questions below.

**Questions For Authors:**

1. If there are no numerical values in the problem (such as in a symbolic computation problem or when the numbers are represented in text), how should the dimension 6 work? and what will the query look like?
2. This paper reveals the performance of each LLM across different dimensions and stages. Based on these findings, the authors highlight potential directions for future improvements in LLMs. Therefore, how do the authors suggest further optimizing each LLM, such as GPT-4?

**Relation To Broader Scientific Literature:**

Although there exist many other studies for LLM evaluation and some dimensions proposed in this paper share similarities, I believe that the organization of the three stages and nine dimensions from a human perspective is both innovative and insightful. It not only clarifies how LLMs achieve performance in each cognitive stage/dimension but also highlights areas for further improvement toward human-like intelligence. As a result, although the conclusion that LLMs are overestimated may not be surprising, the contributions of this paper remain significant for the community.

**Theoretical Claims:**

The core of this paper is to propose an LLM evaluation framework based on human cognition, through which various cognitive issues in LLMs are identified. Therefore, the focus and contribution of this work lie in experimental findings rather than theoretical analysis.

---

> ### Author Rebuttal · Authors · 2025-03-30
>
> We sincerely appreciate your recognition of the psychological foundation of our evaluation framework, the extensive and significant experiments, the valuable insights provided by our work, and the well-organization of our paper.
>
> $\bf{Q1}$: If there are no numerical values in the problem (such as in a symbolic computation problem or when the numbers are represented in text), how should the dimension 6 work? and what will the query look like?
>
> $\bf{A1}$ : Thanks for your detailed review and valuable question, and we apologize for causing this confusion. If the problem does not contain numerical values, then our multi-agent system will not generate any inquiries for Dimension 6, as the Inquiry agent cannot produce any reasonable transformation for the problem. As a result, in such cases, as we outline in Appendix C, we omit consideration of that dimension during the evaluation. For example, the problem "If $a<b$, what is the value of $|a-b|+a+b$?" from MATH dataset does not include any numerical values, so it is unnecessary to assess an LLM's ability of numerical calculations on it. In our framework, we can imagine that our multi-agent systems will also fail to provide a reasonable numerical transformation to such a problem. As a result, we would not evaluate Dimension 6 for this problem, which aligns with real-world scenarios and intuition.
>
> Based on your suggestion, we will add more clarifications in Section 3.2 to better explain our evaluation process.
>
> $\bf{Q2}$: This paper highlights potential directions for future improvements in LLMs. Therefore, how do the authors suggest further optimizing each LLM, such as GPT-4?
>
> $\bf{A2}$ : Thanks for your constructive question. Specifically, we can analyze potential optimization strategies for different LLMs based on the conclusions drawn in Sections 4.3 and 4.4.
>
> For example, as shown in Table 2, weaker LLMs (e.g., Llama2-13B) exhibit the lowest Pass Rates in Stage 1 (i.e., Problem Comprehension), whereas more advanced models (e.g., GPT-4, DeepSeek-V2.5) demonstrate relatively stable comprehension abilities but struggle significantly with mastering Stage 2 (i.e., Problem Solving). This suggests that improving comprehension should be a primary focus for weaker models. Further analyzing Figure 2, we observe that Llama2-13B’s main issues in Stage 1 stem from Dimension 1 (Sentence Paraphrasing) and Dimension 4 (Redundant Condition). This insight suggests that incorporating more training data involving synonymous rewrites and redundant conditions could help enhance its comprehension capabilities.
>
> For GPT-4, its main deficiencies in Stage 1 lie in Dimension 2 (Sentence Disruption) and Dimension 3 (Missing Condition), while in Stage 2, it performs poorly in Dimension 7 (Knowledge Redefinition). For Dimensions 2 and 3, as analyzed in Section 4.4, GPT-4's struggles may stem from its tendency to inherently "over-correct" the unsolvable problems into solvable ones. To address this, we could empower it with the critical thinking skills by reinforcement learning techniques, encouraging the model to recognize counterfactual scenarios instead of merely simulating the reasoning process based on a given input. The results in Dimension 7 indicate that GPT4 may tend to treat knowledge as fixed memorization rather than a flexible reasoning process. This suggests the need for more adaptive knowledge learning strategies, such as contrastive learning and retrieval-augmented training, where models are exposed to dynamic and evolving knowledge sources to encourage reasoning beyond static memorization.
>
> Following your comments, we will supplement these discussions to expand the scope of our work in the revised version.
>
> $\bf{Q3}$: Some Typos.
>
> $\bf{A3}$ : Thanks for your meticulous review and pointing out these typos. We will carefully correct them in the revised version.

---

> > ### Comment · Reviewer_5FjB · 2025-04-04
> >
> > I confirm that I have read the author's response. I will maintain my score.

---

> > > ### Author Response · Authors · 2025-04-04
> > >
> > > Thank you very much for taking the time to read our response! We truly appreciate your valuable comments and will include all our discussions into the revised revision. Thank you again for your thoughtful feedback!

---

### Official Review · Reviewer_wh6Z · 2025-03-14

**Overall Recommendation:** 4

**Summary:**

This paper proposes CogMath to assess LLMs’ abilities at specific cognitive stages. Based on psychological research, they decompose the mathematical reasoning process into three stages: problem understanding, problem solving, and solution summarization. Then, in each stage, they further specify several detailed dimensions to test LLMs’ performance. More specifically, in each dimension, the authors design an innovative “Inquiry-Judge-Reference” multi-agent system to conduct a scientific and reliable evaluation. In experiments, they apply CogMath to several mainstream LLMs. The results reveal that the mathematical abilities of these LLMs are overestimated, and they have distinct advantages and disadvantages at different stages, providing a valuable direction for their further improvement.

**Claims And Evidence:**

Yes. In the proposed CogMath framework, the motivation for decomposing the reasoning process into three stages comes from psychological research. Moreover, the "Inquiry-Judge-Reference" multi-agent system ensures the quality of the "queries-answers" in each dimension. Therefore, the experimental results and findings are convincing.

**Essential References Not Discussed:**

I think the related works are discussed sufficiently.

**Experimental Designs Or Analyses:**

I have checked all the experimental designs, including performance analyses at both the stage and dimension levels, the impact of LLM enhancement methods (e.g., CoT and ICL), error analyses based on difficulty and problem length, and human verification of the multi-agent system. The experiments are well-founded, and their results clearly validate the authors’ conclusions and highlight the contributions of this paper.

**Methods And Evaluation Criteria:**

Yes. As mentioned above, the proposed framework has a solid psychological foundation, and the evaluation dimensions are well-suited to the problem. Besides, the proposed multi-agent system is both reasonable and effective in achieving the goals of this paper.

**Other Comments Or Suggestions:**

Line 18, "assess" -> "assesses"

In this paper, "a LLM" -> "an LLM"

In Figure 2, what is "MathExam"? Does it mean "MExam"?

**Other Strengths And Weaknesses:**

Strengths:

- The paper is well written, with each part detailed enough for understanding. Besides, the authors give clear explanations for their motivation, methodology, and experiments. Therefore, this paper is easy to reproduce and implement.

- The perspective to evaluate LLMs from human cognition is valuable and innovative. It provides a new angle to examine LLMs’ advantages and disadvantages. Besides, the proposed framework has the generalization ability.

- As revealed in Section 4.2, simply introducing more test problems may be insufficient to assess the true mathematical abilities of LLMs. Therefore, this paper points out an important finding that existing related works may overlook.

- The experimental results reveal how different LLMs perform at each cognitive stage and dimension. The exploration of the LLM enhancement methods is a plus for the experimental findings. It provides a new way to assess LLMs’ intelligence and advancement.


There remain some weak issues that can be further improved
- The decomposition of the reasoning process into three stages could be elaborated further.
- The relationships among the proposed dimensions could be discussed in more detail.

**Questions For Authors:**

1. In dimension 2, the authors disrupt the word order within each clause. Why do they choose the clause level, rather than directly disrupt the whole problem?

2. When humans prove theorems, do the three stages discussed in this paper enough to represent the whole reasoning process?

3. As for the nine dimensions in this paper, the authors seem to consider them independent. Do they try to combine some of them in one query and test LLMs?

**Relation To Broader Scientific Literature:**

I believe that the evaluation of LLMs has attracted great attention in current community. There is much literature that intends to evaluate LLMs from different perspectives. However, as the authors state in this paper, these related works either are task-specific or rely on a single accuracy metric to assess LLMs. Therefore, I believe that this paper contributes a valuable angle to view LLMs’ abilities from a human cognitive perspective, with multiple dimensions in each cognitive stage to construct a comprehensive evaluation. Moreover, the proposed framework is more than just a benchmark; it provides a general methodology for evaluating LLM performance across different cognitive stages, even when applied to datasets from various domains. Thus, this framework is extendable.

**Theoretical Claims:**

As this paper focuses on an LLM evaluation framework, there is no issue with theoretical claims.

---

> ### Author Rebuttal · Authors · 2025-03-30
>
> We sincerely appreciate your recognition of our motivation and solid psychological foundation, the credibility of our experimental results, the significant contributions of our framework, and the good writing of our paper.
>
> $\bf{Q1}$: The decomposition of the reasoning process into three stages could be elaborated further. When humans prove theorems, do the three stages discussed in this paper enough to represent the whole reasoning process?
>
> $\bf{A1}$: Thanks for your constructive suggestion and insightful question. Our decomposition of the reasoning process into three stages is based on psychological research, which points out that humans typically undergo three general stages when reasoning about a mathematical problem: Problem Comprehension, Problem Solving, and Solution Summarization. Building on this, we carefully analyze the objectives associated with each stage in mathematical reasoning and designed corresponding dimensions for evaluation.
>
> Moreover, these three stages represent a general evaluation framework. As discussed in Appendix E, they do not rely on specific problem types or formats and can apply to evaluting LLMs' abilities in a wide range of tasks. For instance, in theorem proving you mentioned, the reasoning process of human learners can also be split into understanding the given conditions (corresponding to our Stage 1), performing the proof (Stage 2), and summarizing the proof (Stage 3). Of course, for specific tasks like theorem proving, additional evaluation dimensions could be introduced. For example, in Stage 2, besides numerical computation, we could add assessments related to symbolic manipulation skills.
>
> In summary, the three stages we utilize in this paper are widely generalizable. Depending on the task at hand, we can also introduce more granular dimensions to conduct more evaluation. Based on your suggestion, we will include these discussions in the revised version.
>
> $\bf{Q2}$: The relationships among the proposed dimensions could be discussed in more detail. The authors seem to consider them independent. Do they try to combine some of them in one query and test LLMs?
>
> $\bf{A2}$ : Thanks for your valuable question. Indeed, in our paper, the design of the different dimensions is relatively independent. However, as mentioned in Section 3, only when a LLM passes all dimensions can we conclude that it has genuinely mastered the problem $P$. Therefore, these dimensions are collectively used to assess whether the model truly masters mathematical reasoning.
>
> We certainly appreciate your suggestion to combine different dimensions in a single query, which could provide us with valuable insights into the model's ability to handle multiple dimensions simultaneously. However, this approach may present challenges in assessment in the cognitive stage level, and it could be difficult to trace which specific dimension the model is struggling with. For this reason, we have chosen to evaluate the dimensions independently in our primary experiments.
>
> However, we believe that your suggestion is very promising, and we are willing to explore this direction in future work.
>
> $\bf{Q3}$: Why do they choose the clause level, rather than directly disrupt the whole problem?
>
> $\bf{A3}$ : Thanks for your thoughtful question. We agree that disrupting the whole problem is also a possible approach. In this paper, we chose to disrupt the problem at the clause level because it allows our inquiries to maintain a closer similarity to the original problem's structure while ensuring that they remain unsolvable. As a result, if the model does rely on semantics for reasoning, our approach makes it easier to observe such behaviors. Based on your concern, we will add further clarification about it in the revised version.
>
> $\bf{Q4}$: Typos and does "MathExam" mean "MExam" in Figure 2?
>
> $\bf{A4}$ : Thanks for your meticulous review and pointing out these typos. We will carefully correct them in the revised version.

---

### Official Review · Reviewer_wFDY · 2025-03-23

**Overall Recommendation:** 3

**Summary:**

The paper proposes CogMath, a novel evaluation framework for assessing the mathematical reasoning abilities of LLMs from a human cognitive perspective. Traditional benchmarks primarily focus on answer accuracy, often overestimating LLMs’ true mathematical competence. Instead, CogMath structures evaluation into three cognitive stages—problem comprehension, problem solving, and solution summarization—with nine finer grained evaluation dimensions.  To conduct rigorous assessments, the framework employs an “Inquiry-Judge-Reference” multi-agent system, ensuring models demonstrate a deeper understanding beyond superficial pattern recognition. Applying CogMath to three mathematical benchmarks (GSM8K, MATH, and MExam) reveals that current LLMs' mathematical abilities are overestimated by 30%-40%, with deficiencies varying across the nine dimensions. Additionally, popular prompting techniques like Chain-of-Thought (CoT) and In-Context Learning (ICL) do not significantly enhance genuine reasoning ability. This highlightes the need for more wholistic evaluation of LLM capabilities.

**Claims And Evidence:**

1.) The paper's main claim that due to a single-dimensional nature of existing evaluations, the mathematical reasoning abilities of models are overestimated is well substantiated by the fact that these LLMs when evaluated on a wider range of criteria show significant drops in performance.

2.) In lines 177-178 (right column), the authors state that if humans have mastered a problem (conceptually), changing the numerical values should not affect their ability to solve it. However, intuitively it seems that the change in complexity of computations resulting from the change in numerical values may plausibly affect the ability of arriving at the correct final answer successfully. This argument is even stronger in the case of LLMs which have been shown to struggle with fundamental calculations. It would be important to include evidence from previous literature, if any to support the claim. Absence of it renders Dimension 6 potentially meaningless.

**Essential References Not Discussed:**

Here is a list of references (although not all being "essential") would be good to include

* Any literature related to the claim that human and LLM problem solving abilities are not affected by change in the computations involved (related to dimension 6)
* [1], [2] and [3] related to data contamination

[1] Zhang et al., 2024; A Careful examination of Large Language Model Grade School Performance on High School Arithmetic
[2] Mirzadeh et al., 2024; GSM-Symbolic: Understanding the Limitations of Mathematical Reasoning in Large Language Models
[3] Shah et al., 2024; AI-Assisted Generation of Difficult Math Questions

**Experimental Designs Or Analyses:**

The authors present a several experiments in order to demonstrate the usefulness of the frame work including stage wise as well as dimension wise break down of the performance of different LLMs. The experiments on the effect of Chain-of-Thought and In-Context Learning on the performance of models on difference difficulty levels provides interesting insights.

An important detail however, which seems to be missing from the text is the lenght thresholds corresponding to Length 1 to Length 5.

**Methods And Evaluation Criteria:**

Dimension 6 assumes that the problem solving abilities of LLMs would not be affected by the change in the calculations involved in the problems. This is not true as it has been shown that LLMs struggle at some of the most fundamental computations.

Additionally, I am skeptical about the strategy used in Dimension 2. Depending on how extreme the random shuffling of the sequence of words in the original question is, the result may not assess memorization accurately (if that is the purpose of evaluation along this dimension). For eg. if all the words of the original seuquence are jumbled, the LLM may not be able to solve it correctly even if has partly memorized the original problem.

The evaluation criteria (Pass Rate) seems well thought of for most of the part. However, if the purpose of both Dimensions 2 ans 3 is not to solely evaluate memorization, the strategy of considering any answer except the original answer does not make sense. The LLM arriving at an incorrect but valid numerical answer can point to many other problems apart from memorization (such as hallucination) which also tie in to mathematical capabilities of LLMs.

**Other Comments Or Suggestions:**

The overall writing of the paper could be significantly improved. Below are some suggestions:

* The captions of the figures could be more descriptive and self-containing
* The paper contains several typos. Some typos which I noticed are: Line 49 right column: rom -> from, Line 395 (right): we focusing -> we focus
* A brief description of the three stages (around lines 46-47 (right)) as described in the psychology literature would be helpful.
* I would recommend explicitly stating the purpose and motivation of evaluation along each dimension; i.e. what aspect of mathematical reasoning is aimed to be targeted by each dimension as well its foundation in the psychology literature.

**Other Strengths And Weaknesses:**

I have discussed most of the strengths and weaknesses in other sections. I discuss some additional ones below

### Strengths
* The idea of forming a more wholistic evaluation of LLMs based on psychological performance analogs in humans is a novel and interesting ideas.

### Weaknesses
* Close to zero discussion of the new MExam benchmark introduced in the paper is present in the text. Including some minimal but critical information about the benchmark in the main paper would be important

**Questions For Authors:**

* Are both dimensions 2 and 3 specifically geared towards evaluating memorization of the training data only? If so, it seems a bit redundant (ignoring the concern that Dimension 2 may not be reliably assessing memorization as explained in the "Methodology and Evaluation Criteria" section). Both the dimensions could be be leveraged to attack other kinds of problems in the mathematical capabilities of LLMs (such as hallucination).

**Relation To Broader Scientific Literature:**

This study falls within the broader literature of evaluation of mathematical reasoning in LLMs by addressing the limitations of traditional accuracy-based benchmarks like GSM8K and MATH, instead proposing a cognitively inspired framework, CogMath. It aligns with cognitive science theories on problem-solving and AI evaluation methodologies. The work is also connected to recent research on AI self-verification ([4], [5]), highlighting LLMs' struggles with backward reasoning. While CogMath offers a novel multi-stage evaluation, it could benefit from integrating findings on structured reasoning methods like Tree-of-Thought or Graph-of-Thought, further bridging AI robustness and human-like problem-solving strategies.

[4] Weng et al., 2023; Large Language Models are Better Reasoners with Self-Verification
[5] Yu et al.; 2023; Bootstrap your own mathematical questions for Large Language Models

**Theoretical Claims:**

The authors make no theoretical claims.

---

> ### Author Rebuttal · Authors · 2025-03-29
>
> We sincerely appreciate your recognition of our framework's novelty, the validity of our conclusions, and the insights of our work.
>
> $\bf{Q1}$: Validity of Dimension 6.
>
> $\bf{A1}$: Thanks for your insightful comments. First, the strategy of changing numbers in problems has been widely adopted in assessing learners' abilities. For example, human educators commonly pose numerical variations in exam questions to assess students[1,2]. Similarly, it is widely employed in evaluating a model's mastery for math problems[3,4]. Thus, our Dimension 6 is meaningful for LLM evaluation.
>
> Second, we do not assume that "LLMs' problem-solving abilities would not be affected by the change in the calculations". Instead, if a model's performance greatly declines after numerical variations, our framework can identify such deficiencies in numerical processing capabilities, which indeed aligns with your opinions that LLMs struggle with fundamental computations.
>
> Third, in Dimension 6, our Inquiry agent does not introduce extreme cases (e.g., a huge number) that might exceed current LLMs’ processing limits (an example is shown in Table 6). Therefore, our dimension will not significantly increase the difficulty of the problem, so we do not expect any noticeable decline in model performance.
>
> [1] How to solve it: A new aspect of mathematical method.
>
> [2] Cognitive diagnostic assessment for education: Theory and applications.
>
> [3] GSM-Plus: A Comprehensive Benchmark for Evaluating the Robustness of LLMs as Mathematical Problem Solvers.
>
> [4] Functional benchmarks for robust evaluation of reasoning performance, and the reasoning gap.
>
> $\bf{Q2}$: If all words are jumbled, Dimension 2 may not assess memorization accurately?
>
> $\bf{A2}$: Thanks for your thoughtful question. Our goal in Dimension 2 is not to evaluate the memorization capacity of LLMs. Instead, we aim to assess whether the LLM is truly reasoning or merely relying on the semantic cues in the wording. After randomly shuffling the sequence of words, humans can naturally recognize that the problem becomes unsolvable. Therefore, if an LLM truly master reasoning, it should also recognize this. This is why in Section 3.1 and in Table 6, we clarify that an LLM is considered to have "passed" Dimension 2 if it successfully recognizes the inquiry as “unsolvable”, rather than whether its answer is correct (please also note that the shuffled problem does not have a well-defined correct answer!).
>
> $\bf{Q3}$: Dimensions 2 and 3 evaluate memorization only? Strategy of considering answer?
>
> $\bf{A3}$: Thanks for your insightful questions. As we clarified in $\bf{A2}$, Dimension 2 is not designed to evaluate the memorization, so do Dimension 3. Our goal is to assess whether the LLM is engaging in genuine reasoning, regardless of whether it is responding with memorized information or other capabilities.
>
> In both dimensions, an LLM is considered to pass if it correctly identifies the given inquiry (i.e., $q_2$, $q_3$) as “unsolvable”, rather than considering any specific answers. Therefore, our Pass Rate criteria refers to the proportion of cases where the model successfully recognizes such ill-posed inquiries.
>
> $\bf{Q4}$: Length thresholds.
>
> $\bf{A4}$: Indeed, we do not set fixed thresholds. As stated in Section 4.6, we divide all problems into five levels using an equal-frequency binning approach. This means that we sort all problems by length and then divide them into five equal-sized groups. This ensures that each group contains sufficient data and avoids introducing any potential biases, allowing for more reliable analysis.
>
> $\bf{Q5}$: Information about MExam.
>
> $\bf{A5}$: Thanks for your valuable question. In this work, our goal is not to create a new dataset, MExam, but rather to utilize it to validate that the overestimation of LLM capabilities is not merely due to data contamination (discussed in Section 4.2). Thus, we explain in Appendix C how we collected MExam and the number of problems it contains.
>
> To address your concern, we conduct additional statistical analyses. Due to space limit, please refer to our response $\bf{A4}$ to reviewer $\bf{Xd5A}$ for details. We will also make MExam publicly available if this paper is accepted.
>
> $\bf{Q6}$: Suggestions on writing.
>
> $\bf{A6}$: Thanks for your constructive feedback. We will incorporate the related studies and refine the writing in the revised version.
>
> $\bf{Q7}$: Ethics review.
>
> $\bf{A7}$: Thanks for your attention to ethics. In this work, we invited human annotators to evaluate the outputs of our Judge agents and Reference agents. As stated in Section 4.7, the evaluation protocol was approved by the Ethics Review Board, and all annotators were informed of data usage. Besides, our templates (Appendix B) do not collect private information. Thus, our study does not raise ethical concerns.
>
> We greatly appreciate your in-depth reviews and hope our explanations addresses your concerns. We will also include them in the revised version.

---

> > ### Comment · Reviewer_wFDY · 2025-04-03
> >
> > I appreciate the author's response to my review.
> > I am convinced about the utility of Dimensions 2, 3 and 6. I am increasing my score.

---

> > > ### Author Response · Authors · 2025-04-03
> > >
> > > We sincerely appreciate your constructive comments and valuable feedback! We will incorporate all our discussions and your suggestions into the revised version. Thank you once again for your time and for increasing your score!

---

### Official Review · Reviewer_Whnd · 2025-03-23

**Overall Recommendation:** 3

**Summary:**

This paper introduces CogMath to assess the authentic mathematical reasoning abilities of LLMs through the lens of human cognition. Specifically, the paper models human mathematical reasoning with three stages and nine dimensions, such as sentence paraphrasing, numerical transformation, and backward reasoning. A system with three agents, including Inquiry, Judge, and Reference, is used in each dimension to ensure accurate assessments. Through experiments with CogMath, this paper reveals that current LLMs’ abilities are overestimated by 30%-40% and that prompting techniques do not fundamentally improve LLMs’ mathematical reasoning ability.

**Claims And Evidence:**

Yes. The evidence provided includes: First, there are large-scale evaluations on GSM8K, MATH, and MExam datasets, with 7 mainstream LLMs. Second, the authors provide fine-grained pass rate analysis at different reasoning stages and dimensions, offering clear and significant evaluation results. Third, they also conduct human validation of the multi-agent system to ensure assessment quality.

**Essential References Not Discussed:**

No, the references are essential to understanding this paper.

**Experimental Designs Or Analyses:**

I have checked the experimental designs in Sections 4.1, 4.5, and 4.6. They are reasonable, with evaluations across: three diverse math datasets, seven different LLMs covering both open-source and closed-source, and Dimension-wise analysis to pinpoint LLMs’ weaknesses in different cognitive processes.

**Methods And Evaluation Criteria:**

The three-stage reasoning model aligns with established studies on human problem-solving and the hypothesis that true problem mastery requires passing all nine dimensions is reasonable. Besides, this paper uses Pass Rate for evaluation across different dimensions, which considers both the reasoning scenario and the counterfactual scenario (e.g., Dimensions 2 and 3). Therefore, this evaluation criteria also makes sense.

**Other Comments Or Suggestions:**

N/A

**Other Strengths And Weaknesses:**

In this paper, the authors propose the CogMath framework to evaluate LLMs’ abilities from a cognitive perspective and then conduct sufficient experiments with several LLMs and datasets.

By structuring the evaluation around cognitive stages and dimensions, this paper offers a more fine-grained understanding of LLMs and reveals some interesting phenomena (e.g., the “overcorrect” behavior). Second, the use of a multi-agent system allows for a more rigorous and systematic evaluation process, reducing the limitations of single-metric evaluations and providing deeper insights into reasoning failures. Moreover, this paper covers a diverse set of LLMs and mathematical benchmarks, making the findings broadly applicable.

My major concern about this paper is that although the three-stage reasoning process proposed in the paper is very general, some specialized types of mathematical problems may require additional stages or dimensions to capture the full reasoning process in those areas. Explaining how to expand the framework to address more problem types would further enhance this paper’s applicability. Besides, several newly released advanced LLMs have emerged (e.g., Deepseek-R1, the authors investigate Deepseek-v2.5 in this paper). I recommend the authors to supplement more evaluations on these models.

Additionally, I have a question regarding the multi-agent system. In section 3, the authors state that they set a maximum number of iterations $\delta$ for the agent interaction. However, if after exceeding $\delta$ rounds, the Judge agent still considers the obtained inquiry to be of insufficient quality, how should the evaluation be conducted?

In summary, I think this paper has good quality and good readability, presenting its ideas clearly and logically.

**Questions For Authors:**

1.	I recommend the authors to supplement more evaluations on newly released LLMs (e.g., Deepseek-R1) to analyze their improvements over the previous versions.
2.	If after exceeding $\delta$ rounds, the Judge agent still considers the obtained inquiry to be of insufficient quality, how should the evaluation be conducted?
3.	How is the adaptability of the framework? I would like to hear more discussion about how can the proposed three-stage reasoning process be adapted to more mathematical problem types.
4.	The current training of LLMs largely follows a “pretrain-SFT-RL” process. Given the various weakness identified of different LLMs in this paper, how should we optimize them (e,g, enhance a LLM’s capability in problem comprehension stage) during training?

**Relation To Broader Scientific Literature:**

This paper makes a contribution to the broader scientific literature on evaluating the reasoning abilities of LLMs, particularly in the domain of mathematical problem-solving. While previous benchmarks primarily rely on overall answer accuracy, CogMath provides a cognitively motivated framework that assesses LLMs across multiple dimensions of human-like reasoning. Additionally, it highlights critical weaknesses in existing LLMs, such as the “over-correct” behavior and their struggles with counterfactual reasoning. This study not only provides a more accurate assessment of LLM capabilities but also offers great insights for future model improvements.

**Theoretical Claims:**

The paper focuses on evaluating LLMs through the lens of human cognition and does not involve theoretical issues.

---

> ### Author Rebuttal · Authors · 2025-03-29
>
> We sincerely appreciate your recognition of our framework's soundness, evaluation significance, and great contributions.
>
> $\bf{Q1}$: Adaptability to more problem types.
>
> $\bf{A1}$: Thanks for your insightful question. Our CogMath is easily adaptable because: (1) it is based on the decomposition of human reasoning processes. The three stages, Problem Comprehension, Problem Solving, and Solution Summarization, reflect the general cognitive processes humans use to conduct reasoning. (2) our agents are highly flexible. As shown in Appendix B, their prompts do not impose any specific requirements on the problem types. Besides, their interactions ensure the quality of the generated inquires and reference answers (verified in Section 4.7). (3) our evaluation metrics are reasonable and easy to use, considering both the general "answer correctly" and "unsolvable" in counterfactual scenarios.
>
> $\bf{Q2}$: Evaluation on new LLMs (e.g., DeepSeek-R1).
>
> $\bf{A2}$: Thanks for your constructive suggestion. We supplement results for DeepSeek-R1 below. Due to API speed limitations, here we conduct evaluations on the widely-used public MATH and GSM8K datasets.
>
> | DeepSeek-R1|||||MATH||||GSM8K|
> |-|-|-|-|-|-|-|-|-|-|
> ||Avg|Alg|Count|Geo|Itmd|Num|Pre-Alg|Pre-Cal||
> |Vanilla|0.982|0.992|0.994|0.956|0.979|0.980|0.985|0.972|0.967|
> |CogMath|0.448|0.581|0.443|0.307|0.295|0.413|0.604|0.326|0.703|
> |$\Delta \quad$|-0.534|-0.411|-0.551|-0.649|-0.684|-0.567|-0.381|-0.646|-0.264|
> |$Stage\ 1$|0.863|0.942|0.831|0.737|0.837|0.881|0.875|0.837|0.897|
> |$Stage\ 2$|0.575|0.715|0.557|0.441|0.405|0.544|0.738|0.456|0.848|
> |$Stage\ 3$|0.753|0.808|0.773|0.637|0.694|0.743|0.815|0.725|0.856|
>
> First, DeepSeek-R1 achieves the best performance compared with other LLMs in Table 1, both in “Vanilla” and CogMath framework. This reflects its superior mathematical reasoning capabilities. Second, the performance gap (marked as $\Delta$) suggests that DeepSeek-R1 still exhibits a certain degree of overestimation, highlighting the necessity of our proposed evaluation from the human cognitive perspective. Third, similar to other advanced LLMs, DeepSeek-R1 encounters the most challenges in Stage 2 (Problem Solving). Further investigation reveals that its primary weakness lies in Dimension 7 (Knowledge Redefinition), with a Relative Pass Rate (RPR) of 0.617. This supports the conclusion that current LLMs rely on fixed memorization rather than adapting knowledge flexibly. Lastly, DeepSeek-R1 improves significantly in Stage 3 (Solution Summarization) compared to DeepSeek-V2.5, which suggests a deeper understanding of the reasoning process.
>
> $\bf{Q3}$: How to evaluate after exceeding $\delta$ rounds?
>
> $\bf{A3}$: Thanks for your valuable question. As illustrated in Appendix C, if inquiry quality remains insufficient after $\delta$ rounds, we exclude the dimension because it suggests that the problem may not be suitable for evaluation from that dimension. For example, the problem "If $a<b$, what is the value of $|a-b|+a+b$?" from MATH dataset does not include any numerical values, so it is unnecessary to assess an LLM's ability of numerical calculations on it. Our multi-agent system would also fail to generate a valid transformation, so we would not evaluate Dimension 6, aligning with real-world intuition.
>
> $\bf{Q4}$: How to optimize LLMs (e,g, enhance capability in Poblem Comprehension stage) during "pretrain-SFT-RL" training?
>
> $\bf{A4}$: Thanks for your insightful question. Different phases of training correspond to enhancing different cognitive stages/dimensions of LLMs. For example, the pretraining phase focuses on developing the model's text comprehension abilities and the mastery of fundamental knowledge. The SFT phase is more about teaching the model to simulate a given reasoning strategy. The RL stage allows the model to develop more complex abilities, such as backward reasoning, intermediate step explanations, and error identification.
>
> Therefore, in Problem Comprehension stage, we suggest: 1) For Dimension 1 (Sentence Paraphrasing), increase the corpus used during the pretraining phase. 2) For Dimension 4 (Redundant Condition), train the model with question-answer pairs that include redundant information in SFT phase. 3) For Dimensions 2 (Sentence Disruption) and 3 (Missing Condition), to cultivate critical thinking skills, we can allow the model to think more freely and learn to recognize such situations in the RL phase.
>
> Similarly, for the other cognitive stages, improvements can be made through different training processes. For example, knowledge acquisition in the Problem Solving stage can be reinforced during pretraining, while backward reasoning abilities in the Solution Summarization stage are often enhanced through the RL phase. Overall, our framework provides valuable insights into how to optimize  LLMs in different stages.
>
> We greatly appreciate your thought-provoking suggestions and will include these experiments and discussions in the revised version.

---

### Official Review · Reviewer_Xd5A · 2025-03-25

**Overall Recommendation:** 4

**Summary:**

This paper aims to explore and evaluate the mathematical ability of LLMs. The authors propose a novel evaluation framework (CogMath) based on the human psychological design. The workflow examines the LLM’s performance across three stages, including problem comprehension, problem-solving, and solution summarization. The experiments on real benchmarks reveal several constructive findings for different LLMs.

## update after rebuttal
The authors have adequately addressed my concerns; therefore, I am raising my score to Accept.

**Claims And Evidence:**

The main claim regarding the evaluation findings on LLMs’ reasoning abilities is well supported by the extensive experiments on three benchmark datasets (GSM8K, MATH, and MExam). The CogMath shows the consistent overestimation of LLMs' mathematical abilities. The in-depth analyses of 3 stages and 9 reasoning dimensions provide the robust evidence for the claims about the strengthens and limitations of current LLMs.

**Essential References Not Discussed:**

The authors can cite and discuss the following works, “Deepseekmath: Pushing the limits of mathematical reasoning in open language models." arXiv preprint arXiv:2402.03300”, "Deepseek-v3 technical report." arXiv preprint arXiv:2412.19437.

**Experimental Designs Or Analyses:**

The experimental designs and analyses appear sound and reasonable. The paper tests seven mainstream LLMs on three representative benchmark datasets, ensuring a broad evaluation of their mathematical abilities. The in-depth analyses on nine ability dimensions with the pass rate metric is appropriate for this context. Moreover, the data and method settings are clearly provided in the paper.

**Methods And Evaluation Criteria:**

The proposed methods and its evaluation criteria make sense for understanding and studying the problem. CogMath introduces an "Inquiry-Judge-Reference" multi-agent system to generate multiple inquiries for assessing LLMs’ math ability rather than just testing them with one question. The evaluation criteria on 9 ability dimensions with the proposed metric is appropriate for ensuring the credibility of the results.

**Other Comments Or Suggestions:**

1.Adding statistical analyses on the datasets used in the proposed evaluation.

2.Adding the references with more recent LLMs and provide discussions.

**Other Strengths And Weaknesses:**

Strengths:

1.The paper proposes a novel evaluation framework CogMath for testing LLMs’ reasoning abilities on math problems. The core idea of the framework design aligns the psychological perspectives, which constructs the 3 cognitive stages and 9 ability dimensions. The key technique contribution refers to constructing the multi-agent system to generate multiple inquiries for assessing LLMs’ true ability, rather than just testing them with one question as many existing works do. The overall framework provides comprehensive methods for LLM evaluation.

2.The paper provides an in-depth analysis of the performance of seven mainstream LLMs across different problem types and formats. The CogMath shows the consistent overestimation findings of LLMs on math abilities compared to only using traditional predefined benchmarks, which offers valuable results into their capabilities and limitations.

3.The datasets released for the evaluation can contribute to more study as they involve more diverse questions with the ability labels. That has potentials to further study on LLM reasoning ability exploration and strategy design.

Weaknesses:

1.The paper could include and discuss more recent released deepseek version into the evaluation framework.

2.The scalability of the evaluation framework should be discussed.

3.The pass rate criteria used for 9 dimension seems to have different definitions. In Table 6, I notice the indicators of “Pass” refer to “answer correctly” and “unsolvable”. I cannot find the explicit definitions in the paper. It should be provided

**Questions For Authors:**

1.Can you include and discuss more recent released deepseek version into the in-depth analyses?

2.I am missing the definition of “Pass” criteria in Table 6. How can we distinguish the different indicators in different dimension?

3.Can you discuss the scalability of the proposed framework?

**Relation To Broader Scientific Literature:**

The paper's key contributions are closely related to the broader scientific literature on LLMs and mathematical reasoning. It introduces a novel and comprehensive evaluation framework with a multi-agent system following the workflow of mimicking human reasoning activities of problem comprehension, problem solving, and solution summarization. The findings about the overestimation of LLMs' abilities and the limitations of the mainstream prompting techniques contribute to the existing conclusions. They also have the potential to inspire further research on LLM reasoning ability exploration and strategy design.

**Theoretical Claims:**

The main claims of the paper are evaluated on extensive experiments and qualitative analysis on different datasets, rather than proof. The proposed CogMath introduces a delicate evaluation workflow based on psychological design for the experimental analyses.

---

> ### Author Rebuttal · Authors · 2025-03-29
>
> We sincerely appreciate your affirmation of our framework's novelty, experimental validity, and evaluation significance.
>
> $\bf{Q1}$: Could include more deepseek version.
>
> $\bf{A1}$: Thanks for your valuable suggestion. We will incorporate more references to DeepSeek-related papers. Besides, we conduct experiments on DeepSeek-R1 as follows. Due to API speed limitations, here we conduct evaluations on the widely-used public MATH and GSM8K datasets.
>
> |DeepSeek-R1|||||MATH||||GSM8K|
> |-|-|-|-|-|-|-|-|-|-|
> ||Avg|Alg|Count|Geo|Itmd|Num|Pre-Alg|Pre-Cal||
> |Vanilla|0.982|0.992|0.994|0.956|0.979|0.980|0.985|0.972|0.967|
> |CogMath|0.448|0.581|0.443|0.307|0.295|0.413|0.604|0.326|0.703|
> |$\Delta \quad$|-0.534|-0.411|-0.551|-0.649|-0.684|-0.567|-0.381|-0.646|-0.264|
> |$Stage\ 1$|0.863|0.942|0.831|0.737|0.837|0.881|0.875|0.837|0.897|
> |$Stage\ 2$|0.575|0.715|0.557|0.441|0.405|0.544|0.738|0.456|0.848|
> |$Stage\ 3$|0.753|0.808|0.773|0.637|0.694|0.743|0.815|0.725|0.856|
>
> First, compared to Table 1, DeepSeek-R1 achieves the best performance among all evaluated LLMs, both in “Vanilla” and CogMath framework. This shows its superior mathematical reasoning capabilities. Second, based on the performance gap (marked as $\Delta$), DeepSeek-R1 still exhibits a certain degree of overestimation, highlighting the necessity of our proposed evaluation from the human cognitive perspective. Third, similar to other advanced LLMs, DeepSeek-R1 encounters the most challenges in Stage 2 (i.e., Problem Solving). Further analysis reveals that its main weakness lies in Dimension 7 (Knowledge Redefinition), with a Relative Pass Rate (RPR) of 0.617. This supports the conclusion that current LLMs rely on fixed memorization rather than adapting knowledge flexibly. Fourth, compared to DeepSeek-V2.5, DeepSeek-R1 improves significantly in Stage 3 (Solution Summarization), suggesting a deeper understanding of reasoning process.
>
> $\bf{Q2}$: Scalability of CogMath.
>
> $\bf{A2}$: Thanks for your constructive comments. CogMath is highly scalable because: (1) it is designed based on human reasoning processes, which makes the three stages independent of any specific problem types. (2) our multi-agent system is highly flexible. As shown in Appendix B, they do not depend on the dataset or task definition. Besides, their interactions ensures the quality of our inquiries and reference answers (verified in Section 4.7). (3) our evaluation metric is widely applicable, considering both general “answer correctly” and “unsolvable” in counterfactual situations.
>
> $\bf{Q3}$: Definitions of Pass Rate.
>
> $\bf{A3}$: Thanks for your valuable question. As explained in Section 4.1, Pass Rate for Dimensions 1 and 4-9 refers to the accuracy of answering the inquiries correctly. For Dimensions 2 and 3, they are counterfactual evaluation dimensions. For example, Dimension 2 evaluates the LLM’s response after the words of the original problem are randomly shuffled (an example is in Table 6). Ideally, this will render the problem meaningless, and the LLM should not provide the original answer. Thus, in this case, Pass Rate refers to the proportion of cases where the LLM successfully identifies the inquiry as "unsolvable".
>
> $\bf{Q4}$: Statistical analyses on datasets.
>
> $\bf{A4}$: Thanks for your constructive suggestion. We present the number of problems (#P), the average problem length (Avg.P), and the average answer length (Avg.A) of the original dataset, along with the average inquiry length (Avg.$q_i$) and the average answer length (Avg.$a_i$) for each dimension $i$ in our framework. Since the reference answers $a_i$ for Dimensions 1 to 4 are same to the original answers and the inquiry $q_2$ for Dimension 2 is simply a shuffled version of the original problem, no additional statistics are required for these cases.
>
> ||MATH|GSM8K|MExam
> |-|-|-|-|
> #P|5000|1319|6353
> Avg.P|29.51|46.91|133.81
> Avg.A|81.70|49.13|115.62
> Avg.$q_1$|33.25|46.30|132.07
> Avg.$q_3$|24.18|38.20|128.51
> Avg.$q_4$|47.28|63.52|235.71
> Avg.$q_5$|28.20|47.38|140.25
> Avg.$a_5$|133.20|61.67|200.64
> Avg.$q_6$|30.49|46.82|148.70
> Avg.$a_6$|199.99|106.24|180.59
> Avg.$q_7$|51.76|70.69|178.20
> Avg.$a_7$|266.01|201.88|580.72
> Avg.$q_8$|15.82|12.80|40.60
> Avg.$a_8$|215.46|111.23|402.66
> Avg.$q_9$|47.56|66.47|113.74
> Avg.$a_9$|1.12|1.01|1.15
>
> We observe that the inquiries in Dimensions 4, 7, and 8 exhibit the most significant length differences compared to the original problems. This is expected, as they ntroduce additional conditions, redefine knowledge concepts, or only ask about one specific intermediate step. Furthermore, in most cases, the reference answers are longer than the original answers. Upon further inspection, we find that this is not due to an increase in problem difficulty, but rather stems from our Reference agent providing a more detailed solution, whereas the original dataset answers are more concise.
>
> We sincerely appreciate your thoughtful comments and will incorporate these experiments and discussions in the revised version.

---

### Decision · Program_Chairs · 2025-05-01

**Decision:**

Accept (poster)

**Comment:**

This paper proposes a new framework to evaluate the authentic mathematical reasoning ability of large language models. CogMath breaks down evaluation into three psychological stages: problem comprehension, problem solving, and solution summarization. The perspective to evaluate LLMs from human cognition is valuable and innovative. It provides an in-depth analysis of the performance of seven mainstream LLMs across different problem types and formats. Additionally, Inquiry-Judge-Reference agents ensure robust and unbiased evaluation. Potential disadvantages include the complexity of implementation, potential redundancy in dimensions, and limited discussion of the MExam dataset.